# Musculoskeletal Disorder Symptoms among Workers at an Informal Electronic-Waste Recycling Site in Agbogbloshie, Ghana

**DOI:** 10.3390/ijerph18042055

**Published:** 2021-02-19

**Authors:** Augustine A. Acquah, Clive D’Souza, Bernard J. Martin, John Arko-Mensah, Duah Dwomoh, Afua Asabea Amoabeng Nti, Lawrencia Kwarteng, Sylvia A. Takyi, Niladri Basu, Isabella A. Quakyi, Thomas G. Robins, Julius N. Fobil

**Affiliations:** 1Department of Biological Environmental and Occupational Health Sciences, School of Public Health, University of Ghana, Accra 00233, Ghana; jarko-mensah@ug.edu.gh (J.A.-M.); duahdwomoh@yahoo.com (D.D.); effya76@gmail.com (A.A.A.N.); lawkwarps@yahoo.com (L.K.); satakyi002@st.ug.edu.gh (S.A.T.); profquakyi@gmail.com (I.A.Q.); jfobil@ug.edu.gh (J.N.F.); 2Center for Ergonomics, Department of Industrial and Operations Engineering, University of Michigan, Ann Arbor, MI 48109-2117, USA; crdsouza@umich.edu (C.D.); martinbj@umich.edu (B.J.M.); 3Faculty of Agricultural and Environmental Sciences, McGill University, Montréal, QC H9X 3V9, Canada; niladri.basu@mcgill.ca; 4Department of Environmental Health Sciences, School of Public Health, University of Michigan, Ann Arbor, MI 48109-2029, USA; trobins@umich.edu

**Keywords:** musculoskeletal disorders, ergonomics, electrical and electronic waste (e-waste), informal work, Agbogbloshie

## Abstract

Informal recycling of electrical and electronic waste (e-waste) has myriad environmental and occupational health consequences, though information about the chronic musculoskeletal health effects on workers is limited. The aim of this study was to examine the prevalence and intensity of self-reported musculoskeletal disorder (MSD) symptoms among e-waste workers at Agbogbloshie in Ghana—the largest informal e-waste dumpsite in West Africa—relative to workers not engaged in e-waste recycling. A standardized musculoskeletal discomfort questionnaire was administered to 176 e-waste workers (73 collectors, 82 dismantlers, and 21 burners) and 41 workers in a reference group. The number of body parts with musculoskeletal discomfort were 1.62 and 1.39 times higher for collectors and dismantlers than burners, respectively. A 1-week discomfort prevalence was highest for collectors (91.8%) followed by dismantlers (89%), burners (81%), and the reference group (70.7%). The discomfort prevalence for e-waste workers was highest in the lower back (65.9%), shoulders (37.5%), and knees (37.5%). Whole-body pain scores (mean ± SE) were higher for collectors (83.7 ± 10.6) than dismantlers (45.5 ± 7.6), burners (34.0 ± 9.1), and the reference group (26.4 ± 5.9). Differences in prevalence, location, and intensity of MSD symptoms by the e-waste job category suggest specific work-related morbidity. Symptom prevalence and intensity call attention to the high risk for MSDs and work disability among informal e-waste workers, particularly collectors and dismantlers.

## 1. Introduction

### 1.1. The Nature of the E-Waste Problem

The fervent demand for new technology coupled with the fast obsolescence and short lifespan of modern-day consumer devices has created a global crisis in electrical and electronic waste (e-waste) recycling [1,2,3,4]. The term e-waste describes all types of discarded electrical and electronic equipment/appliances and its parts (e.g., televisions, laptops, refrigerators, automotive sub-assemblies). The global volume of e-waste in 2019 approximated 53.6 m metric tons and is projected to exceed 74 m tons by 2030, i.e., a growth rate of 2 m tons per year [5]. Most e-waste is generated from domestic consumption [5,6]. In addition, large volumes of e-waste from Europe and North America are shipped each year to countries in Asia [7,8], South America [6,9], and Africa [10,11,12]. E-waste shipped to developing countries include legal exports intended for low-cost recycling, illegal dumping that evade national and international laws, and items sent under the guise of donations [5,13,14]. A small fraction of all shipped e-waste items are in working condition and get put to second-hand use. However, the majority (over 80%) consists of non-working goods that end up in dumpsites to be either dismantled for parts which are refurbished and resold, or recycled into scrap material [10]. E-waste recycling in developing countries is dominated by the informal sector comprising low skilled workers who manually collect, dismantle, and burn/melt e-waste as a source of income [5]. E-waste is hazardous since many components contain hazardous chemicals and heavy metals (e.g., lead, cadmium) harmful to the environment and to human health if not handled and/or disposed of safely [15,16]. The environmental and occupational health risks associated with e-waste recycling activities present unique challenges in developing countries due to the lack of appropriate e-waste recycling infrastructure, policy, and governance [2,10,17,18].

This study focuses on Agbogbloshie in Accra, Ghana, the largest dumping grounds for e-waste in Africa [16]. It is among the largest and busiest informal e-waste recycling sites in the world [19]. Over 215,000 tons of e-waste was imported into Ghana in 2009 [13]. The Agbogbloshie scrapyard processes between 10,000 to 13,000 metric tons of e-waste annually [20]. Conditions at the e-waste site have made Agbogbloshie one of the most polluted places on earth [12,21].

E-waste recycling work performed at Agbogbloshie is unregulated and informal (i.e., without any formal organizational structure, work procedures) with the primary goal of recovering re-usable parts, isolating precious metals and other scrap material for sale [22,23]. Workers at the site engage in multiple manual tasks, though depending on the primary work tasks performed they broadly fall into three categories, collectors, dismantlers, and burners [22,24]. E-waste collectors typically travel offsite within the nearby communities by foot or bicycle to search, purchase, and collect e-waste items, and transport these back to the worksite [22,25]. Dismantlers manually disassemble irreparable and/or non-functional e-waste items using their bare hands and rudimentary tools such as a chisel, hammer, and pliers to salvage reusable components [22]. Burners perform open burning of e-waste items, including insulated electrical cables, automotive wire harnesses, and other components that cannot be further dissembled in order to isolate valuable metals (e.g., gold, copper, iron, and aluminium). E-waste workers at this site are low skilled, mostly migrants and farmers from the northern part of Ghana that travel south in search of work [26,27]. They have little education or awareness of occupational health and safety such as the use of personal protective equipment (PPE; e.g., gloves, safety shoes, safety glasses, dust masks) or ergonomics, nor the means to buy PPE or professional work-tools [22,26]. E-waste workers at Agbogbloshie are among the poorest and most vulnerable members of the urban populations in Ghana [23,28].

### 1.2. Environmental and Occupational Health Effects

E-waste work at Agbogbloshie has gained substantial research attention due to the scale of operations and associated health effects on workers and the local community. Adverse health effects include neurological and genetic disorders [19,27,29,30,31] and respiratory issues [23,32,33,34]. A major source of exposure stems from open-air burning of insulated components that release toxic chemicals (e.g., brominated flame retardants) and heavy metals (e.g., lead, mercury, cadmium) into the ambient air, soil, and water [35,36,37,38]. These permeate the food and water supply, and eventually the bodies of workers and nearby residents [19,27,31]. Elevated noise exposures and potential noise-induced hearing loss are also a concern [23,33,39,40,41].

### 1.3. Work-Related Musculoskeletal Disorders

Informal e-waste recycling is manual and labour-intensive [3]. It involves long durations of sitting, standing, walking, sitting in a low squatting posture, with frequent and heavy manual material handling (MMH) in non-neutral postures (e.g., lifting, carrying, pushing-pulling) [22,25]. The equipment used for transporting (e.g., hand-drawn carts, wheelbarrows), dismantling (e.g., hammers, chisels, pliers), and burning (i.e., long metal rods for handling burning items) is simple and worn out [22]. Only 25% of e-waste workers wear any PPE such as safety shoes and gloves [41]. These conditions present a potential risk for both, acute injuries and chronic work-related musculoskeletal disorders (WRMSDs). Work-related acute injuries, e.g., cuts, bruises/contusions and burns, are common among e-waste workers [11,22,41,42].

In contrast, systematic evaluations of MSDs from e-waste work are limited. WRMSDs include sprains, strains, tears, carpal tunnel syndrome, and connective tissue disorders that manifest as localized chronic inflammation, pain, soreness, and/or discomfort in the affected body part. Rather than any single causal event, WRMSDs result from long-term exposure to ergonomic risk factors including non-neutral (awkward) postures, highly repetitive movements, forceful exertions, contact stress, and vibration coupled with limited periods of rest and recovery [43,44,45,46]. Much of the causal evidence on WRMSDs is based on epidemiological research conducted in industrial settings [43,44,47]. MSDs are the most common work-related health problem globally placing a substantial economic burden on employers, employees, and the health system [48,49,50,51].

Prior studies conducted at Agbogbloshie provide some indication of WRMSD symptoms among e-waste workers. These include descriptive accounts of chronic body pain and discomfort and an overuse of pain medications [22,23,34]. A recent survey of 84 e-waste workers at Agbogbloshie by Fischer et al. found a high prevalence of back pain (88% vs. 69.9% among non-e-waste workers) and neck pain (44.6% vs. 43% among non-e-waste workers) [41]. Studies on e-waste workers in other countries suggest similar trends [11,42,52]. These prior studies focused only on a few and notably different body parts with no data on musculoskeletal discomfort in other body parts, which limit a full understanding of the potential risk for developing MSDs. The lack of a standardized questionnaire methodology and resulting diversity in prevalence periods and symptom definitions hampers comparisons of MSD symptom prevalence across studies. Ergonomics research, on the other hand, has relied on standardized musculoskeletal discomfort questionnaires to screen for WRMSDs [53,54,55,56,57,58]. The use of such validated questionnaires enable comparisons of MSD symptoms across countries, cultures, work settings, and time characteristics.

The aim of this study was to assess the prevalence and intensity of WRMSD symptoms among informal e-waste workers at the Agbogbloshie recycling site using the standardized Cornell Musculoskeletal Discomfort Questionnaire (CMDQ) [54]. The study hypothesized a higher frequency (i.e., 7-day period prevalence) and intensity (i.e., using weighted pain scores) of self-reported musculoskeletal discomfort among the three e-waste worker categories (collectors, dismantlers, and burners) compared to a reference group of workers not engaged in informal e-waste recycling.

## 2. Materials and Methods

This study used a cross-sectional design and was conducted at two locations: The e-waste recycling site at Agbogbloshie in Accra, Ghana, and at Madina Zongo, a suburb of greater Accra, Ghana. Data collection was part of the GEOHealth-II project and occurred between January to March 2018 [59]. The ethical and protocol review committee of the University of Ghana College of Health Sciences approved the study. Participant recruitment at both locations relied on a convenience sampling of workers that were present, interested, and available on the day of data collection. All the study participants provided written informed consent. Consent for participants younger than 18 years was obtained from their immediate work supervisor who were typically their parent, relative or guardian.

### 2.1. Study Site and Sample

Initially, a community durbar (i.e., public gathering that often includes noblemen) was held at Agbogbloshie including the local Chiefs of Agbogbloshie and Madina Zongo, leaders of the scrap dealers association, e-waste workers available at the time of the durbar, and members of the research team. The durbar presented an opportunity to explain the study objectives and procedures, obtain buy-in from local leaders and stakeholders, and recruit interested e-waste workers.

The Agbogbloshie e-waste site is located near the center of Accra, the capital of Ghana. The site is approximately 0.5 km^2^ and bordered by the Korle lagoon, the Odaw river, and a local food market [12,25]. By some estimates, nearly 5000 workers show up at the Agbogbloshie e-waste site every day [5]. In order to diversify the sample in terms of job category, interested and available e-waste workers were recruited from different locations of the worksite after the study purpose had been explained. The study recruited 176 e-waste workers, including 73 collectors, 82 dismantlers, and 21 burners.

A reference group of workers that did not engage in e-waste recycling activities were recruited from Madina Zongo located about 10 km to the north of Agbogbloshie. Individuals in the reference group had similar ethnic background as the e-waste workers at Agbogbloshie and were mostly from northern Ghana. The reference group consisted of a convenience sample of 41 participants from diverse occupations, i.e., shop attendants (n = 8), traders (n = 20), vehicle drivers (n = 4), students (n = 3), and a couple each of schoolteachers, tailors, and unemployed youth, many of whom did not typically perform heavy force exertions or MMH as part of their routine work.

### 2.2. Study Procedure

After explaining the study procedures and receiving consent, all participants were administered a questionnaire to obtain information on demographics (e.g., age, gender) and occupation, including their primary job category (e.g., collector, dismantler, or burner for e-waste workers), years worked in the current job, typical number of days worked per week, and the number of hours worked per day. Occasionally, an e-waste worker would associate their work tasks with more than one e-waste job category. In such cases, follow-up questions were asked emphasizing tasks performed most often in the prior workweek in order to record the primary and secondary job categories that the participant self-identified with. Since such cases were few and their counts reported (i.e., thirteen collectors and dismantlers, three collectors and burners, four dismantlers and burners), only the primary job category was used for the purposes of statistical data analysis.

Next, all the participants were administered the CMDQ, which is detailed in the subsequent section. Completion of the CMDQ required approximately 10 min. Data collection including administering of questionnaires were performed at both Agbogbloshie and Madina Zongo concurrently by four different researchers who were all trained by the principal investigator (A.A.A.) in the study procedures including the use of the CMDQ. The questionnaires were administered in English, and when needed, explanations were given in Dagbani, the local dialect spoken by e-waste workers. Participants’ verbal responses were recorded on paper, and later coded into an electronic spreadsheet for analysis.

### 2.3. Cornell Musculoskeletal Discomfort Questionnaire

The CMDQ was used to obtain information about MSD symptoms (e.g., discomfort, aches, and pains) experienced in the previous 7-day workweek [54,60,61]. Adapted from the Nordic Musculoskeletal Questionnaire [56], the CMDQ is a widely used screening tool for musculoskeletal discomfort complaints with established psychometric properties applied to diverse occupations [54,62,63,64,65] and different language translations [66,67,68]. The questionnaire provides a standardized methodology to screen for musculoskeletal discomfort in 18 major body parts, namely, the neck, shoulders (2), upper arms (2), forearms (2), wrists/hands (2), upper back, lower back, hip/buttocks (2), thighs (2), knees (2), and lower legs including ankles/feet (2), along with a procedure for computing an aggregate symptom score for the whole body. A body-map diagram accompanies the questionnaire depicting the body parts where the individual may have experienced MSD symptoms in the prior workweek. Participants were instructed to use the full workweek prior to when the questionnaire was presented as the reference period (i.e., a 7-day period starting Monday morning).

The questionnaire assesses musculoskeletal discomfort at each body part in three sections, *frequency*, *severity*, and *work interference*. *Frequency* corresponds to the number of times discomfort, aches or pain was experienced in the past 7 days, and categorised as either none, 1–2 days/week, 3–4 days/week, once every day, or several times a day. Specific to the present study, participants who reported experiencing discomfort for more than 4 days in the past week (e.g., on 5 or 6 workdays) were instructed to select their frequency response as either “once daily” or “several times a day” based on the respective within-day frequency. *Severity* and *work interference* ratings for each body part are obtained only if the corresponding frequency rating exceeded “none”. *Severity* of the reported musculoskeletal symptom is assessed by a rating of either slight, moderate, or very uncomfortable. *Work interference* assesses whether musculoskeletal complaints interfere with work by a rating of not at all, slightly, or substantially.

In order to identify body parts with the most serious MSD symptoms, the CMDQ provides a procedure for obtaining an aggregate score (which we term “pain score” to differentiate from the term “discomfort rating”) using pre-defined weights applied to the participants’ ratings [54,61]. Pain scores for each body part were computed by multiplying the weighted frequency (0, 1.5, 3.5, 5.0, 10.0), severity (1.0, 2.0, 3.0), and work interference (1.0, 2.0, 3.0) ratings, respectively. In essence, pain scores convert the three ordinal rating scales into a single continuous measure ranging from 0 to 90. To facilitate the comparisons of pain scores across job categories, the 18 body parts were combined into four body regions, i.e., lower extremities (sum of both knees, lower legs, thighs, and hips/buttocks), upper extremities (sum of both shoulders, upper arms, forearms, and wrists/hands), lower back, and the upper back and neck. Whole-body pain scores were computed as the sum of pain scores for all four body regions.

### 2.4. Statistical Analysis

Statistical analyses focused on examining differences in demographics and musculoskeletal discomfort among the four job categories, i.e., collectors, dismantlers, burners, and the reference group. All of the statistical analyses were conducted using IBM SPSS v. 24 (IBM Corp). Summary statistics for age, years on the job, work hours per day, and number of days worked per week were tabulated and compared across job categories using separate one-way ANOVA tests (and a non-parametric Kruskal-Wallis test for “work days per week”). Tests yielding a significant difference among job categories (*p* < 0.05) were followed by pairwise comparison tests with the significance value adjusted for multiple comparisons using the Bonferroni-correction. All pairwise combinations in job categories were included in the post hoc analysis.

Completed discomfort questionnaires were transcribed and imported into SPSS for statistical analysis. Counts for discomfort frequency, severity, and work interference ratings were tabulated by job category (Appendix A, Appendix A) and subjected to three types of analyses. First, the number of body parts (i.e., out of 18 total) per participant with a discomfort frequency rating of once or more in the prior workweek were computed. A single Poisson regression was performed to estimate the number of body parts with discomfort as a function of job category, and adjusting for age, years of work, number of days worked per week, and the number of work hours per day included as continuous covariates (*p* < 0.05).

Second, discomfort prevalence (%) for each body part was computed as the proportion of participants by job category that reported a discomfort frequency rating of once or more in the prior workweek. Separate Chi-square tests of proportions for each body part were performed to compare the discomfort prevalence (%) among job categories (*p* < 0.05). Significant effects of job category were further analysed using pairwise Chi-square tests with a Bonferroni adjustment for multiple comparisons (*p* < 0.05).

Third, pain scores (i.e., the product of weighted frequency, severity, and work interference ratings) were computed and summarized by job category. The distribution of pain scores for the whole body and four body regions were skewed left and not normally distributed, hence five separate non-parametric Kruskal-Wallis tests were used to examine differences in pain scores by job category. Significant effects of job category (*p* < 0.05) were subjected to Kruskal-Wallis pairwise comparisons with the significance value adjusted for multiple tests using the Bonferroni correction (*p* < 0.05).

## 3. Results

### 3.1. Demographic and Occupational Characteristics of the Study Sample

The study sample consisted of 217 male participants including 73 (33.6%) collectors, 82 (37.8%) dismantlers, 21 (9.7%) burners, and 41 (18.9%) in the reference group. Table 1 summarizes the age, years on the job, work hours per day, and number of days worked per week stratified by the four worker categories. Participant ages ranged from 11 to 55 years, with six collectors and one reference group member being minors (<18 years old). In general, e-waste workers were significantly younger than the reference group (*p* < 0.05) by an average of 5 to 8 years depending on the particular e-waste worker category (Table 1). Among e-waste worker categories, collectors were significantly younger than dismantlers by an average ± standard error (SE) of 2.9 ± 1.1 years (*p* = 0.049).

The years worked on the job ranged from 1 week to 30 years. In general, e-waste workers had worked for fewer years on the job compared to the reference group. Dismantlers and the reference group had worked significantly more years on the job compared to collectors, with a mean ± SE difference of 2.5 ± 0.9 years (*p* = 0.046) and 3.4 ± 1.2 years (*p* = 0.025), respectively. Participants worked for an average ± SE of 9.9 ± 0.2 h per day and for a median ± inter-quartile range (IQR) of 6 ± 1 days per week. The hours worked per day and days worked per week were very diverse ranging from 1 to 7 days a week and between 1 to 15 h per day depending on the type of work and availability of work each day, but did not differ significantly among the job categories (Table 1).

### 3.2. Number of Body Parts with Discomfort

The median ± IQR number of body parts with discomfort were higher for collectors (4 ± 5) followed by dismantlers (3 ± 4), then burners (2 ± 3), and the reference group (2 ± 5). Results from the Poisson regression performed on the number of body parts with discomfort indicated a statistically significant effect of job category, age, and hours worked per day, but no significant effect of years on the job nor days worked per week (Table 2).

Compared to the reference group, the number of body parts with discomfort reported was significantly higher for collectors by 1.62 (95% CI, 1.25–2.10) times and for dismantlers by 1.39 (95% CI, 1.08–1.79) times. The number of body parts with discomfort reported by dismantlers was 0.86 (95% CI, 0.73–1.0; *χ*^2^ = 3.65, *p* = 0.056) times lower but not significantly different compared to collectors. The number of affected body parts reported by burners were 0.62 (95% CI, 0.46–0.82; *χ*^2^ = 10.948, *p* = 0.001) times lower compared to collectors, and 0.72 (95% CI, 0.54–0.96; *χ*^2^ = 4.882, *p* = 0.027) times lower compared to dismantlers. Differences in the number of affected body parts between burners and the reference group were not significant (*p* = 0.985). A 1-year increase in age was associated with a 0.98 (95% CI, 0.97–1.0) times decrease in the number of body parts from which discomfort is perceived (Table 2). Every additional hour of work each day was significantly associated with a 1.07 (95% CI, 1.03–1.10) times increase in the number of body parts with discomfort (Table 2).

### 3.3. Prevalence of Musculoskeletal Discomfort

Figure 1 provides a graphical summary of discomfort prevalence by body part and job category. Prevalence statistics for the right and left sides of the upper and lower extremities were similar. Hence, only data for the right side are presented. Comparing across body parts, discomfort prevalence for e-waste workers (vs. the reference group) was highest in the lower back (65.9% vs. 51.2%), followed by the shoulders (37.5% vs. 31.7%), knees (37.5% vs. 19.5%), lower legs (26.7% vs. 14.6%), upper arms (28.4% vs. 2.4%), and neck (26.1% vs. 22.0%). Chi-square tests of proportions indicated statistically significant differences in discomfort prevalence by job category for the knees (*p* = 0.001), lower legs (*p* < 0.001), and upper arms (*p* < 0.001), with the higher discomfort prevalence among collectors driving most of these differences. Discomfort prevalence for the remaining body parts did not differ significantly among the job categories.

### 3.4. Pain Scores by Body Region and Job Category

The weighted frequency, severity, and interference scores were multiplied to obtain a weighted pain score for each body part. Table 3 summarizes the means, medians, and ranges for the pain scores aggregated into four main body regions and a whole-body score. Figure 2 provides a graphical high-level comparison of the average pain scores by body region and job category. The cumulative pain score (mean ± SE) for the whole body was the highest for collectors (83.7 ± 10.6) followed by dismantlers (45.5 ± 7.6), burners (34.0 ± 9.1), and the reference group (26.4 ± 5.9), respectively, with pain scores for the lower extremities (i.e., knees, lower legs, thighs, hip/buttocks) and upper extremities (i.e., shoulders, upper arms, forearms, wrists) being the major contributors. Whole-body pain scores were significantly higher for collectors compared to dismantlers and the reference group, but not burners (Table 3). Pain scores for the lower extremities were significantly higher for collectors compared to dismantlers, burners, and the reference group. Pain scores for the upper extremity were comparable between collectors and dismantlers and significantly higher compared to the reference group but not burners. As a single body part, the lower back alone had an average pain score ranging from 13.4 ± 1.9 for collectors to 7.2 ± 2.0 for the reference group. However, the differences among the job categories was marginally non-significant (*p* = 0.081). Pain scores for the upper back and neck combined were of smaller magnitude compared to the other three body regions, and did not differ by job category (*p* = 0.563).

## 4. Discussion

The present study compared WRMSD symptoms of discomfort, aches, and pain among three worker categories (i.e., collectors, dismantlers, and burners) at an informal e-waste recycling site at Agbogbloshie, Ghana and a reference group of workers not engaged in e-waste recycling. Notably, our findings indicate that the number of body parts affected, their location, frequency (i.e., 7-day period prevalence), and intensity (i.e., weighted pain scores) of musculoskeletal discomfort was different between the four worker categories suggesting specific work-related morbidity. Detailed comparisons of musculoskeletal discomfort between the major body parts affected and between e-waste worker categories were possible through the CMDQ. This contrasts with the few previous studies on musculoskeletal discomfort prevalence among informal e-waste workers that were limited to 2–3 specific body parts with non-standard questionnaires [11,41].

### 4.1. Musculoskeletal Discomfort and Disorder Risk

Our findings on discomfort prevalence, the number of body parts affected, and pain scores suggest that the risk of WRMSDs and work disability is higher for collectors and dismantlers than burners and the reference group. A 1-week discomfort prevalence was highest for collectors (91.8 ± 3.2%) and dismantlers (89.0 ± 3.5%) followed by burners (81.0 ± 8.6%) and lastly, the reference group (70.7 ± 7.1%) though the latter was non-trivial. The number of body parts affected was 1.624 to 1.392 times greater for collectors and dismantlers, respectively, relative to burners and the reference group. The location of body parts contributing to the high discomfort prevalence for collectors and dismantlers was also informative (Figure 1). Body parts with a high prevalence of musculoskeletal discomfort for collectors and dismantlers included the lower back (67 vs. 68%), shoulders (36 vs. 42%), upper arms (30 vs. 33%), and neck (25 vs. 28%). Collectors additionally reported a high discomfort prevalence in the knees (52 vs. 28%) and lower legs including ankles/feet (47 vs. 13%) that together point to a high risk of lower extremity MSDs, while the rate of wrist and hand discomfort was twice higher for dismantlers (15%) compared to collectors (7%). Discomfort prevalence for burners was comparatively less and with the lower back (52%), shoulder (29%), knees (24%), neck (24%), and wrists/hands (15%) being most influential.

These findings on musculoskeletal discomfort prevalence corroborate concerns of high prevalence of back, neck, and shoulder discomfort among informal e-waste workers at Agbogbloshie [41], though substantially higher than the 1-year prevalence rates for e-waste workers in Nigeria [11]. The differences in prevalence periods and questionnaire wording suggest caution when comparing prevalence rates across different studies [69]. Based on a survey of 84 e-waste workers at Agbogbloshie, Fisher et al. reported an 88% prevalence of back pain, which presumably includes the upper and lower back combined [41]. Stratified by job category, prevalence rates were 100% for collectors, 92.3% for dismantlers, 85% for burners, and 69.9% in the reference group [41]. The corresponding 7-day discomfort prevalence for the low back in our study were comparatively lower at 67% for collectors, 68% for dismantlers, 52% for burners, and 51% for the reference group, but substantially higher than the 29% 1-year prevalence of low back pain for e-waste workers in Nigeria [11]. The emphasis on low back disorders in prior studies is not surprising. Low back disorders are a common global health problem and a leading cause of work disability [70]. A systematic review of 165 population studies across occupations and countries indicated an average ± SD point prevalence and 1-month prevalence for low back pain to be 18.3 ± 11.7% and 30.8 ± 12.7%, respectively [71]. The prevalence for low back discomfort among informal e-waste workers including those from our study are substantially higher than these global estimates.

Our study also indicated high rates of shoulder discomfort ranging between 29–42% on average across e-waste job categories compared to the 14% prevalence among e-waste workers in Nigeria [11]. For the neck, we found an average discomfort prevalence between 22–28% among the four worker categories, which was lower than the 44.6% prevalence (vs. 43% in the reference group) reported by Fisher et al. [41], but higher than the 10% 1-year prevalence reported by Ohajinwa et al. [11]. However, neither of the latter studies provided a breakdown by job category.

High rates of MSD symptoms, particularly in the low back, shoulders, neck, and knees plague the informal waste processing sector more broadly [72,73]. Back pain was the most common health complaint (67.2%) among municipal waste collectors in Germany (n = 65), while the prevalence of musculoskeletal complaints in other body parts was low at 15.4% [74]. A survey of 340 solid waste collectors (e.g., sweepers, garbage collectors, garbage van, and tricycle drivers) in Accra, Ghana found high pain prevalence in the back (73.5%), wrist (48.2%), and neck (44.7%) with 80 to 90 % of the respondents developing pain symptoms subsequent to joining their current job [75]. A survey of 220 municipal solid waste collectors in India indicated a 1-week and 1-year MSD symptom prevalence of 91.8% and 70%, respectively with the most affected body parts consisting of the knees (84.5% 1-week prevalence), shoulders (74.5%), and lower back (50.9%) [76]. Abou-Elwafa et al. reported a higher 1-year overall prevalence of musculoskeletal complaints among solid waste collectors in Egypt at 60.8% (compared to 43.6% in a comparison group), particularly in the low back (22.5%) and shoulders (15.8%) and to a lesser extent in the neck, knees, hips/thighs, and elbows (<10% each) [77]. Studies from Colombia, Philippines, and Iran also report a high prevalence of MSD symptoms among urban waste workers [78,79,80,81]. These studies also discuss potential risk factors including heavy MMH (e.g., lifting, carrying, pushing-pulling), bending and repetitive tasks, and prolonged sitting and standing, environmental factors (e.g., heat, harsh outdoor conditions), personal factors (e.g., low education levels, smoking, drug use), and psychosocial factors (e.g., work stress, poor job security, lack of organizational support) [72]. Collectively, these studies call attention to the musculoskeletal health effects of informal and unstructured work encountered in the manual processing/recycling of municipal solid waste, medical waste, and e-waste—the latter being the emphasis of this study.

### 4.2. Influence of E-Waste Work Tasks

Pain scores (based on a maximum score of 90.0) analysed in our study provide insight into the relative intensity of MSD symptoms among job categories. Whole-body pain scores (mean ± SE) for collectors (83.7 ± 10.6) were about twice more compared to dismantlers (45.5 ± 7.6) and burners (34.0 ± 9.1). Despite the similar discomfort prevalence between collectors and dismantlers (i.e., 91.8 vs. 89.0%), the substantially higher pain scores for collectors suggest a higher MSD symptom severity and interference with work (indicating potential work disability) compared to corresponding ratings for dismantlers. The lower extremities (38.7 ± 50.7) and upper extremities (24.9 ± 33.8) were major contributors to the high whole-body pain scores for collectors, while the upper extremities was the main contributor to whole-body pain scores for dismantlers (18.5 ± 33.2) and burners (14.5 ± 24.1). The low back consistently contributed an average of 13.4 to 9.0 points towards whole-body pain scores for all three e-waste categories.

Trends in the prevalence, location, and intensity of MSD symptoms potentially reflect the specific work activities performed by collectors and dismantlers such as prolonged walking, standing, floor/low-level sitting, and heavy MMH reported in prior studies of e-waste work at Agbogbloshie [22,23,34,39]. The use of rudimentary hand tools for transporting, dismantling, and burning of e-waste coupled with the lack of PPE presents an additional risk of WRMSDs. For instance, studies indicate that collectors spend most of their workday walking through nearby communities searching and collecting e-waste items and transporting them back to the site using hand-drawn carts, in a cloth sack over the shoulder, or occasionally in hired vehicles [22,23]. Thus, MMH activities of pushing and pulling the hand-drawn carts and lifting and carrying items when loading/unloading the cart was common for collectors [22,23]. The weight of the loaded cart potentially differs from day to day as a function of items identified for recycling. Multiple ergonomics studies document an increased risk of work-related shoulder and low back disorders from MMH including lifting, carrying, pushing, and pulling [82,83,84,85]. Both prolonged walking and standing transfer loads from the upper body to the spine contributing to low back pain [86,87,88,89], and to the lower extremities affecting pain/discomfort in the knees, ankles, and feet [90,91]. Roads in and around the Agbogbloshie e-waste site are uneven and unpaved, soft and muddy during rainy periods, and hard and rough during the dry season [23,34]. This further increases stress in the hands, shoulders, and lower back when pushing/pulling hand-carts [45,92], and increases metabolic energy demands, fatigue, and MSD symptoms of pain/discomfort [93,94]. Walking on uneven terrain also increases loading in the muscles and ligaments around the knees and ankles, resulting in an increased risk of MSDs in these joints [95,96]. While disassembling e-waste items, prior studies reported that dismantlers either stood in a bent/stooped posture, or sat on a low stool or on a dismantled appliance such as an old cathode ray tube television or microwave covering in a squatting posture [22]. Prolonged sitting while working in non-neutral postures is associated with chronic low back pain [82,84]. This may also explain the high prevalence of low back pain for the reference group. Prior studies suggest that a squatting posture that regularly involves severe knee and ankle flexion for a cumulative 1 h in an 8 h workday or 5 min/h increases chronic discomfort in the knees and ankles [95,96,97]. Dismantling of e-waste items was done using bare hands and tools such as hammers, chisel, pliers, and screwdrivers, and involved highly repetitive and forceful upper extremity movements [22], which are known risk factors for MSDs in the shoulder, elbow, hands, and wrists [43,97]. Dismantlers typically walk shorter distances compared to collectors, and primarily for transporting dismantled parts such as insulated components, cables, and wires by hand or in a wheelbarrow to the burners for extracting valuable metals, e.g., gold, copper, and aluminium [22,23]. Loads handled by burners are mainly from the weight of items being burnt (e.g., insulated cables, wires). Burners stood for long durations in mild trunk flexion while holding/manipulating a long metal rod to lift and flip wires/cables during the open-air burning process [22]. The prolonged standing with hand loads and extended upper body postures are potential factors influencing MSD symptoms in the upper and lower extremities and low back.

### 4.3. Sampling Considerations

Beyond the physical work demands, certain demographic and socioeconomic characteristics specific to informal e-waste workers might suggest an increased risk of WRMSDs and subsequent disability (i.e., their inability to continue physical tasks both, at work or outside of work). The e-waste workers at this site were relatively young (mean age ± SE of 24.7 ± 0.5 years vs. 31.2 ± 1.4 in the reference group) with relatively few years of experience in e-waste recycling at an average ± SE of 6.3 ± 0.4 years, which may lower expectations of MSD symptom prevalence compared to the reference group. However, workers in this study on average worked over 8 h a day and over 6 days a week, conditions that point to reduced opportunity for rest/recovery, increased cumulative fatigue, and higher MSD risk. E-waste recycling at Agbogbloshie continues every day of the week, with workers taking either Fridays or Sundays off based on their religious affiliation [23]. Prior reports have also indicated that most e-waste workers in Ghana are young (typically between 14 to 40 years old), regularly work between 10 to 12 h per day or 300 to 360 h per month, and have a short work tenure (i.e., high employment turnover) of 3 to 7 years [13,20]. E-waste workers at Agbogbloshie are mostly migrants from the northern part of Ghana seeking low-skilled jobs, and a portion of them return back after a few years of e-waste work due to work-related injuries and disability [98]. The significant but small negative association between age and the number of body parts affected with MSD symptoms (Table 2) might suggest a healthy worker effect, whereby older and more affected workers were either absent or quit e-waste work. However, confirming this would require a prospective study compared to the present cross-sectional study design.

All the study participants self-identified as male corroborating prior reports that men mostly perform the e-waste recycling work at Agbogbloshie [12,19,23]. The high physical demands of the job is a potential reason for the informal e-waste recycling being male dominant [22,23]. Women preferred less strenuous supportive roles such as vending food and water to workers, or selling water to burners for cooling extracted metal after burning in lieu of manually intensive e-waste recycling [25,99]. Future studies could consider expanding the scope to include the health effects of e-waste work on women in these supporting roles. Akormedi et al. reported that beginner workers at Agbogbloshie, mostly migrants, start out as e-waste collectors and later transition to more skilled jobs such as dismantling [23]. Our sample partially corroborates this since collectors were significantly younger (mean diff. ± SE: 2.9 ± 1.1 years; *p* = 0.049) and had worked fewer years (2.5 ± 0.9 years; *p* = 0.046) in e-waste recycling compared to dismantlers, though comparisons with burners were not significantly different. In addition, about 8% (6/73) of the collectors in our study were minors who typically performed entry-level tasks such as assisting older/senior workers with e-waste collection until they acquired experience and capital to work independently. Child labour is not unique to Agbogbloshie. In Ghana, 21% of children between the ages of 5 to 17 years are child labourers, and 14% of children are engaged in hazardous forms of labour [100]. Many developing countries grapple with child labour in the informal waste collection/recycling sector including informal e-waste recycling in China [101,102]. These trends might suggest deficiencies in the enforcement of legislation and policies to curb child labour [18,100].

In summary, the high rates of MSD symptoms that suggest an increased risk of MSDs and work disability are potentially a confluence of multiple factors. These include the high physical work demands, a relatively young worker population, long work hours, low literacy (e.g., health risk awareness, safe work practices), poor work conditions (e.g., polluted environment, lack of proper PPE, and work tools), and psychosocial stressors associated with informal e-waste recycling. The latter includes heightened job stress due to psychological demands, poor social support, work-related violence and harassment, financial difficulties from low income, low sense of agency in influencing their work situation, and limited opportunity for other forms of gainful employment [22,23,34,39]. The local scrap dealers association at Agbogbloshie does not maintain precise records on the number of workers nor does it provide any regulatory control over work activities undertaken by individual workers at the site—which only emphasizes the informal and unstructured nature of e-waste recycling encountered at Agbogbloshie, Ghana and at similar informal e-waste sites in many developing countries.

### 4.4. Study Limitations

This study used a cross-sectional survey design as a first step to understanding the location, frequency, and intensity of self-reported MSD symptoms among different categories of e-waste workers relative to a group of non-e-waste workers. Certain methodological limitations stem from this choice of study design. Foremost is that a cross-sectional design does not provide causal relationships between specific work exposures and MSDs, and would require a prospective study design. However, such studies need extensive resources and personnel.

The WRMSD symptoms documented in this study relied on retrospective self-reports. Self-report questionnaires are widely used as a reliable and cost- and time-efficient method to screen for MSD symptoms in ergonomics studies [58]. While retrospective self-reports of pain/discomfort are subject to inaccuracies and bias, the possibility would be lower for short vs. longer periods of recall (e.g., 1-week in this study vs. 1-year or 1-month recall). Subsequent studies could incorporate clinical evaluations of MSDs—however, such assessments rely on medical professionals and require substantial financial and personnel resources. These ensuing studies could optimize their limited resources by focusing assessments on specific combinations of body parts and worker categories with high prevalence and/or pain scores which were identified in the present study.

Bias in workers responses to the questions could stem from self-perceptions on their work activity exposures, a low sense of agency in terms of improving their work conditions, low literacy, nuances in local dialects, and differences in the interpretation or comprehension of the questionnaire [22,26]. To minimize these influences, the present study employed professional research staff familiar with English and the local language for administering the questionnaire and recording responses. The research staff received training and conducted pilot studies with the CMDQ to ensure measurement consistency. E-waste workers at Agbogbloshie are also known to self-medicate on painkillers and traditional medications to treat body aches and pain [22,26]. A survey of 84 e-waste workers at this site found high rates of substance abuse (25%; e.g., smoking cannabis) and pain medication (57.5%; e.g., painkillers) [41]. The habitual use or abuse of such treatments could have masked discomfort, aches, and pains associated with underlying WRMSDs among some participants.

The informal, unregulated nature of the worksite and worker population presented challenges to recruiting a representative sample in terms of job category, age, and work experience, which might affect the generalisability of findings. However, the study team made efforts to sample as many workers as possible and from different locations within the e-waste site to ensure an adequate representation of participants. Likewise, attempts were made to recruit a diverse cohort of non-e-waste workers at the comparison site. Burners comprised only 9.7% of the overall study sample. The latter may have reduced statistical power in pairwise comparisons involving burners for some of the outcome measures, e.g., between collectors and burners for whole-body pain scores. The smaller number of burners vs. collectors and dismantlers at Agbogbloshie and in our study sample may be due to a limited need for this job, differences in job content, and/or financial prospects. Burners earn less per day compared to collectors and dismantlers (i.e., USD 16 compared to USD 26 and 52, respectively) [23]. Burning as a primary job function is also less appealing to workers due to the high exposure to smoke and toxic fumes from open burning, and the reliance on dismantlers to provide items for burning [22,23].

Since workers at this site did not have assigned job titles or designations, job categories were derived from self-reported work descriptions. This could cause potential misclassification of the primary job category. Workers at the e-waste recycling site in Ghana are involved in multiple tasks [27,31]. Twenty participants reported a primary and secondary e-waste job. These cases were not differentiated due to their small number and limited influence on MSD symptom prevalence and intensity. However, the issue of workers provisionally changing their primary job tasks on their own over time may have implications for future studies focused on assessing occupational exposures in the informal e-waste worker population.

The reference group and e-waste worker cohort recruited were similar in gender and ethnic composition. Inclusion of the reference group helps provide context to the study findings on MSD symptoms among the e-waste workers. However, workers in the reference group had diverse occupations, many being less physically demanding, and thus different from informal e-waste recycling in terms of physical work exposure (e.g., task type, duration, and intensity). This suggests some caution when drawing conclusions from this study on differences in the prevalence, location, and/or intensity of MSD symptoms between both groups.

### 4.5. Study Implications for MSD Prevention

Despite the above limitations, a strength of the present study was its use of a standardized questionnaire methodology (i.e., CMDQ) to document and compare general complaints of work-related musculoskeletal discomfort in a diverse group of e-waste workers relative to a group of non-e-waste workers. In doing so, our study could improve future comparisons of WRMSD symptoms among e-waste worker populations in different locations and over time characteristics. The present findings could also help inform the targeted design, implementation, and evaluation of evidence-based measures aimed at MSD prevention among the e-waste workers. The unstructured and decentralized nature of informal e-waste processing coupled with the work-related differences in MSD risk reflected in our findings might suggest a multipronged approach to MSD prevention [72]. The hierarchy of controls is a widely adopted approach to organizing different solutions for hazard control and injury prevention [103]. The hierarchy considers hazard elimination and/or substitution the most effective and preventative. The next levels are engineering controls to mitigate hazards between the source and worker, and administrative controls that attempt to alter the way work is performed (e.g., training, team-based work), respectively. The final and least effective level of the hierarchy is PPE use [103]. An example of hazard elimination and/or substitution at the Agbogbloshie site was the introduction of an electric-powered wire-stripper and shredding machine to isolate metal (e.g., copper) from insulated cables and wire, thereby reducing some need for open burning [16,104]. The intervention required a large multi-year investment, an emphasis on training locals about proper use and maintenance, and depended on e-waste burners to fundamentally alter their work behaviours and overcome concerns of lower yield and decreased income—aspects were all contingent on local buy-in and mutual trust [16,104]. Studies on informal e-waste work also focus their recommendations on PPE (e.g., work gloves, face masks, shoes/boots) which work around the hazard, and potentially stall the process at the least effective level of the hierarchy of controls [11,33,41]. Engineering and administrative controls to reduce the continuous physical exposures and associated WRMSD risk among informal e-waste workers remain relatively under-explored, however, require a structured job analysis of exposure to ergonomic stress in the workplace. Studies also point to the need for broader access to healthcare among e-waste workers [26,41]. Reviews on prior interventions in the informal waste recycling sector emphasize the need for participatory approaches to intervention design and implementation that continuously engage (i.e., before, during, and after) and nurture trust with affected workers and community stakeholders [16,72]. The diverse socioeconomic realities and work conditions across countries and locations where informal e-waste recycling is conducted implies that the development and implementation of MSD prevention measures be adapted to the local context.

## 5. Conclusions

This initial study provides evidence on the musculoskeletal health effects of manual e-waste recycling. Overall, the musculoskeletal discomfort frequency (i.e., 1-week period prevalence) and intensity (i.e., weighted pain scores) was higher among e-waste workers compared to the reference group. The MSD symptom prevalence was highest in the lower back, followed by the shoulders, knees, lower legs, upper arms, and neck. The high rates of MSD symptoms identified in a relatively young worker population, alongside the poor work conditions (e.g., polluted environment, lack of basic hand tools, and PPE) and physical work demands suggest an elevated risk of WRMSDs and disability.

Comparing the MSD symptom prevalence, the number of body parts affected, and weighted pain scores by e-waste job category indicated that the risk of WRMSDs and work disability is higher for collectors and dismantlers compared to burners. Differences in the specific e-waste recycling tasks performed potentially help explain these patterns in MSD symptoms among the e-waste workers. Developing and implementing a targeted approach to reduce the risk of chronic and disabling injuries in this population will need a systematic evaluation of the associations between specific e-waste work exposures and WRMSDs. This need is urgent considering the global volume of e-waste generated each year and the large scale of informal recycling conducted in many developing countries [5,10].

## Figures and Tables

**Figure 1 ijerph-18-02055-f001:**
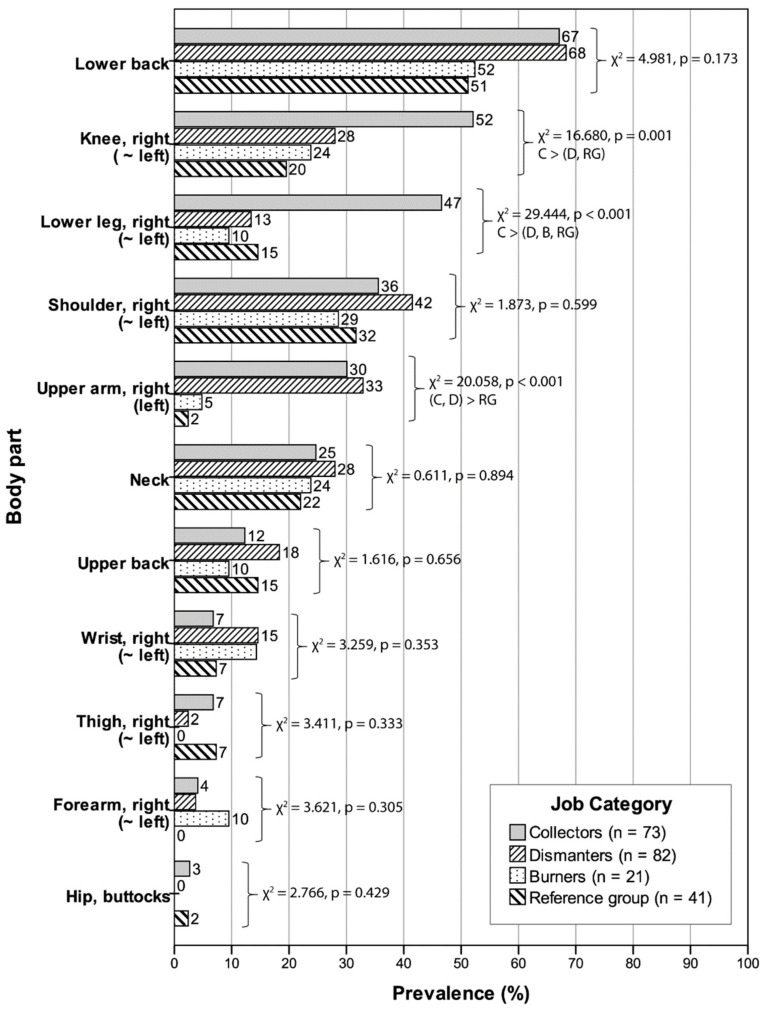
Prevalence (%) of musculoskeletal discomfort for each body part by job category sorted in descending order of discomfort prevalence among the collectors, along with Chi-square test results and Bonferroni-adjusted pairwise comparisons for body parts with significant differences in discomfort prevalence among job categories (*p* < 0.05). Prevalence statistics between bilateral body segments were similar, hence, only data for the right side are presented.

**Figure 2 ijerph-18-02055-f002:**
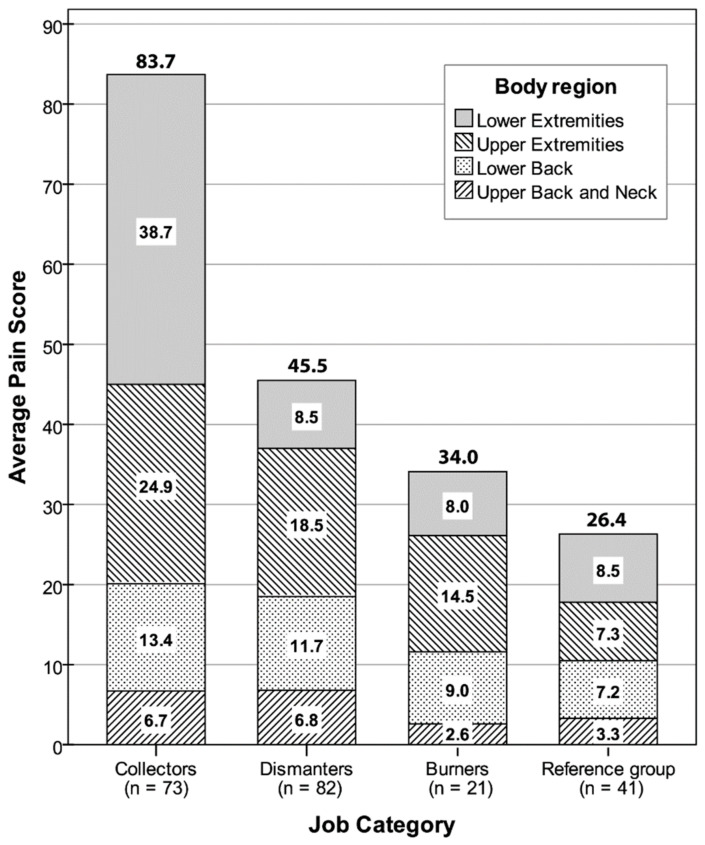
Stacked bar graph depicting average whole-body pain scores by job category. Shaded stacks depict average pain scores for the four body regions: Lower extremities (knees, lower legs, thighs, hip/buttocks), upper extremities (shoulders, upper arms, forearms, wrists), lower back, and upper back and neck.

**Table 1 ijerph-18-02055-t001:** Summary statistics on demographic variables stratified by the four primary job categories. The test statistic summarizes results from separate one-way ANOVA tests (and non-parametric Kruskal-Wallis test for “work days per week”) and Bonferroni-adjusted pairwise comparisons for tests with significant differences among job categories (*p* < 0.05; indicated in bold).

Variable		E-Waste Workers (n = 176)				Test Statistic
	Summary Statistic	Collectors (C; n = 73)	Dismantlers (D; n = 82)	Burners (B; n = 21)	Reference Group (RG; n = 41)	
Age (years)		23.4 ± 0.73	26.4 ± 0.73	22.9 ± 0.88		*F*(3, 204) = 12.6, ***p* < 0.001;** RG > (C, D, B); D > C
	Min–Max	11–43	18–42	18–30	17–55	
Years on the job	Mean ± SE	5.1 ± 0.66	7.6 ± 0.59	5.5 ± 0.68	8.5 ± 1.24	*F*(3, 207) = 4.05, ***p* = 0.008;** (RG, D) > C
	Min–Max	0.02–22.0	1.0–25.0	1.0–13.0	1.0–30.0	
Hours per day	Mean ± SE	10.0 ± 0.26	9.8 ± 0.34	9.5 ± 0.59	10.2 ± 0.47	*F*(3, 210) = 0.325, *p* = 0.807
	Min–Max	1.0–14.0	1.0–14.0	2.0–12.0	2.0–15.0	
Work days per week	Median ± IQR	6 ± 1	6 ± 1	6 ± 1	6 ± 1	*χ*^2^(3, 210) = 1.64, *p* = 0.650
	Min–Max	4–7	2–7	2–7	1–7	

**Table 2 ijerph-18-02055-t002:** Summary results from a Poisson regression predicting the number of body parts with discomfort reported by participants in the past week based on job category, and covariates of age, years on the job, hours worked per day, and days worked per week. Significant effects at *p* < 0.05 are indicated in bold.

Parameter	Estimate, B ± SE	Wald *χ*^2^, *p*-Value	Exp(B)	95% CI
Intercept	1.178 ± 0.338	*χ*^2^ = 12.17, ***p* < 0.001**	3.247	1.676–6.294
Collectors (vs. reference group)	0.485 ± 0.132	*χ*^2^ = 13.54, ***p* < 0.001**	1.624	1.254–2.103
Dismantlers (vs. reference group)	0.331 ± 0.128	*χ*^2^ = 6.66, ***p* = 0.010**	1.392	1.083–1.789
Burners (vs. reference group)	0.003 ± 0.182	*χ*^2^ = 0.0, *p* = 0.985	1.003	0.703–1.433
Age (years)	−0.018 ± 0.008	*χ*^2^ = 6.07, ***p* = 0.014**	0.982	0.967–0.996
Years on the job	0.008 ± 0.009	*χ*^2^ = 0.77, *p* = 0.380	1.008	0.991–1.025
Hours worked per day	0.064 ± 0.015	*χ*^2^ = 17.56, ***p* < 0.001**	1.066	1.034–1.098
Days worked per week	−0.057 ± 0.041	*χ*^2^ = 1.90, *p* = 0.168	0.945	0.871–1.024

**Table 3 ijerph-18-02055-t003:** Mean ± standard error (SE), median ± inter-quartile ranges (IQR), and minimum–maximum ranges of pain scores (max. score 90.0) by job category cumulative for the whole body and for the four main body regions: The lower extremities (knees, lower legs, thighs, hip/buttocks), upper extremities (shoulders, upper arms, forearms, wrists), lower back, and upper back and neck. The test statistic summarizes results from separate non-parametric Kruskal-Wallis tests and Bonferroni-adjusted pairwise comparisons for tests with significant differences among the job categories (*p* < 0.05; indicated in bold).

Body Region	Summary Statistic	Collectors (C; n = 73)	Dismantlers (D; n = 82)	Burners (B; n = 21)	Reference Group (RG; n = 41)	Test Statistic
Whole Body Pain Score (sum of four body regions)	Mean ± SE	83.7 ± 10.6	45.5 ± 7.6	34.0 ± 9.1	26.4 ± 5.9	*χ*^2^ = 20.802, ***p* < 0.001;** C > (D, RG)
	Median ± IQR	45.0 ± 114.75	21.0 ± 43.0	20.5 ± 39.5	7.5 ± 38.5	
	Min–Max	0–360	0–461.5	0–140	0–180	
Pain Score—Lower Extremities	Mean ± SE	38.7 ± 50.7	8.5 ± 21.8	8.0 ± 15.4	8.5 ± 21.7	*χ*^2^ = 26.841, ***p* < 0.001;** C > (D, B, RG)
	Median ± IQR	12.0 ± 71.5	0 ± 9.0	0 ± 15.0	0 ± 6.0	
	Min–Max	0–180	0–150	0–56	0 -120	
Pain Score—Upper Extremities	Mean ± SE	24.9 ± 33.8	18.5 ± 33.2	14.5 ± 24.1	7.3 ± 18.5	*χ*^2^ = 10.673, ***p* = 0.014;** (C, D) > RG
	Median ± IQR	10.0 ± 41.0	6.0 ± 20.0	3.0 ± 19.0	0.0 ± 6.5	
	Min–Max	0–170	0–180	0–90	0–100	
Pain Score—Lower back	Mean ± SE	13.4 ± 1.9	11.7 ± 1.8	9.0 ± 3.4	7.2 ± 2.0	*χ*^2^ = 6.743, *p* = 0.081
	Median ± IQR	6 ± 25.5	5.5 ± 15	1.5 ± 9	1.5 ± 9.5	
	Min–Max	0–45	0–90	0–45	0–60	
Pain Score—Upper Back and Neck	Mean ± SE	6.7 ± 12.7	6.8 ± 12.9	2.6 ± 4.8	3.3 ± 8.3	*χ*^2^ = 2.046, *p* = 0.563
	Median ± IQR	0 ± 8.0	0 ± 10.5	0 ± 2.5	0 ± 2.3	
	Min–Max	0–50	0 -61.5	0–14	0–40	

## Data Availability

Data supporting the reported results are provided under Appendix A for this manuscript.

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
