# Peer review of "Musculoskeletal Disorder Symptoms among Workers at an Informal Electronic-Waste Recycling Site in Agbogbloshie, Ghana"

_ijerph, 2021, doi:10.3390/ijerph18042055_

Round 1

Reviewer 1 Report

The strength of this study is the use of a standardized questionnaire, the sample size and use of a reference population.  Limitations are the cross-sectional design and potential selection bias. It would be of value to readers for the authors to evaluate factors (discomfort level and body region) that are associated with substantial interference with work.

L144    How were the 176 e-waster workers selected/recruited from the 5000 workers?  Were the only job categories at the site: collectors, dismantlers and burners?  Approximately, how many of the 5000 workers who worked at the e-waste site were in each of those categories?

L145    Later we learn that only males are included in the study.  Why?  Did only males work at all sites?

L147    How were the 41 reference workers recruited?

L164    When were the interviews done? Beginning, middle or end of workday or anytime during the workday?

L197    Do prior publications of aggregate CMDQ score use the term ‘pain score’?  If not, you should use the same term used in prior publications.

L201    It is not clear how the summation facilitates comparisons since the number of body parts (18) is different for each of the 4 body regions. Therefore, if pain is the same in each body part, the pain scores will be larger in the regions with the most body parts.

L206    Analysis.  The inclusion of a worker’s body part with “slight” discomfort in the analyses degrades the importance of this report.  Almost everyone has “slight” discomfort throughout the work week – including this reviewer and likely every co-author of this paper.  It would be much more impactful to include only those with moderate or very uncomfortable discomfort/pain and those with substantial interference with work. [And those taking medications for the discomfort, which was not assessed].  This is somewhat dealt in Section 3.4 by reporting the “pain score” that using weighting to discount “slight”.

L212    Was there a formal statistical test for normality of each of the outcome measures?

The paper alternates between using SD and SE.  Please only use SD.

L246    remove “(0.02 years)”

L259    Overall, results are reported with too many significant digits.

L259    It is not clear what the value is of repeating in the text all the numbers that are already in the table.  The text should be reduced.

L284    Fix the insert (Figure 2)

L287    Perhaps in the text just present the findings of significantly different body regions: knee, leg, upper arm.

Figure 1           The order of jobs in the Job Category box should be reversed to match the figure.

L309    Fix the insert.

Figure 2 is useful and the box insert is correctly ordered.

L378    change to “165 population studies”; change SE to SD; change “pain/discomfort” to “pain” to accurately reflect the findings from reference 71.

L380    Ref 71 reviewed studies of low back “pain”.  Your study evaluated low back “ache, pain, discomfort”.  Therefore, one would expect the prevalence in your study to be higher.

L465    The lower age in the e-waste group compared to the reference group is a modifier and one would expect a lower prevalence of discomfort in the e-waste group.  This should be mentioned.

Author Response

Response to Reviewer 1 Comments

Reviewer1-1: The strength of this study is the use of a standardized questionnaire, the sample size and use of a reference population.  Limitations are the cross-sectional design and potential selection bias. It would be of value to readers for the authors to evaluate factors (discomfort level and body region) that are associated with substantial interference with work.

Response1: Thank you for reviewing our work. We agree with the strengths and limitations pointed out by the reviewer. We think we have adequately addressed the limitations of a cross-sectional study design and potential selection bias in Section 4.4. Study Limitations (lines 489 – 537).

We chose a cross-sectional study design as the first step in exploring work-related musculoskeletal disorders among e-waste workers at Agbogbloshie. This is the typical approach for understudied worker populations or work domains. Future studies could include a longitudinal study design to explore in detail the factors that lead to musculoskeletal disorders. It is worth mentioning that we are currently working on a separate ergonomics study to evaluate risk factors associated with the risk of development work-related musculoskeletal disorders among e-waste worker at Agbogbloshie.

Reviewer1-2: L144  How were the 176 e-waster workers selected/recruited from the 5000 workers?  Were the only job categories at the site: collectors, dismantlers and burners?  Approximately, how many of the 5000 workers who worked at the e-waste site were in each of those categories?

Response 2: We appreciate the opportunity to elaborate on these aspects of the study. We address the comment in three parts: number of workers, job categories, sampling.

First, there is no published record of the exact numbers of workers at this site, nor is there data on the number (or proportion) of workers by job category, but rather only approximations. None of the 25+ scientific studies we cited on Agbogbloshie, many of which are cross-sectional in design, provide an exact count nor percentage by job category [16-19, 23-41]. We also mention in the Discussion that “The local scrap dealers association at Agbogbloshie does not maintain precise records on the number of workers nor does it provide any regulatory control over work activities undertaken by individual workers at the site” (lines 484-486). This is a problem inherent to informal work settings. The number “5000” is an estimate of the number of workers at this site in a recent UN report, i.e., “By some estimates, nearly 5,000 workers show up at the Agbogbloshie e-waste site every day” (line 137). This is the only reported number that we came across and neither does it imply that is the total size of the e-waste work population. We mentioned this number only to provide context for the general size/scale of informal e-waste processing at Agbogbloshie. We are quite comfortable leaving out this number since it does not detract from nor diminish the contribution of our study. Counting or estimating the number of workers in each category in an unregulated, informal context would be beyond the scope of the present work, but we share the reviewer’s general sentiment that such information would have been useful, if available.

Second, collectors, dismantlers and burners are the primary job categories directly related to e-waste processing at Agbogbloshie. Prior literature documents this and multiple other studies conducted at this site use a similar if not identical three-group classification. Our own observations conducted at this site also corroborate this. Dismantlers and collectors may be involved in sorting activities for portions of time, but don’t consider it their primary job. There are other allied tasks (e.g., vending food/water by women) but those are not directly about e-waste processing. We understand that the processes and worker categories may be different in other e-waste sites and countries. Hence, we attempted to provide a brief background about the local work context in the Introduction (e.g., lines 62-70) and Discussion, and provide references where available.

Third, this study used a convenience sample method at both study locations, i.e., the Agbogbloshie e-waste site and at Madina Zongo. We had previously indicated that workers available on the day/time of the study and were interested/willing to participate were recruited for the study, as follows “In order to diversify the sample in terms of job category, interested and available e-waste workers were recruited from different locations of the worksite after the study purpose had been explained” (lines 139-141). For additional clarity regarding convenience sampling, we have added the following sentence to the Methods, “Participant recruitment at both locations relied on a convenience sampling of workers that were present, interested and available on the day of data collection” (lines 128-129). We also added the following phrase to the paragraph describing the reference group, “consisted of a convenience sample of 41 participants” (lines 147)

We hope the explanations above address the reviewer’s concerns.

Reviewer1-3: L145    Later we learn that only males are included in the study.  Why?  Did only males work at all sites?

Response 3: Thank you for the opportunity to clarify this point. Yes, e-waste recycling work at this site was only performed by males (i.e., men and minors/boys). Hence, we also recruited male participants in the reference group to avoid confounding by gender. We clarified that the study sample was entirely male in the Results “The study sample consisted of 217 male participants” (line 231), and elaborate in the Discussion section “All study participants self-identified as male corroborating prior reports that men mostly perform the e-waste recycling work at Agbogbloshie [12,19,23]. The high physical demands of the job is a potential reason for the informal e-waste recycling being male dominant [22,23]. Women preferred less strenuous supportive roles such as vending food and water to workers, or selling water to burners for cooling extracted metal after burning in lieu of manually intensive e-waste recycling [25,99].”  We consider the present explanation is sufficient as-is. The tasks performed by women were not directly related to the study aim, i.e., e-waste recycling and hence were not included in the study sample. It is possible that women perform informal e-waste processing in other countries; however, such comparisons would be outside the scope of the present study.

Reviewer1-4: L147    How were the 41 reference workers recruited?

Response 4: Thank you for the comment. Similar to the cohort of e-waste workers, the 41 workers in the reference group were also recruited based on a convenience sample of workers present in the Madina Zonga. This followed a similar methods used in other studies conducted under the GEOHealth-II consortium [41]. For additional clarity regarding convenience sampling, we have added the following sentence to the Methods, “Participant recruitment at both locations relied on a convenience sampling of workers that were present, interested and available on the day of data collection” (lines 128-129).

Reviewer1-5: L164    When were the interviews done? Beginning, middle or end of workday or anytime during the workday?

Response 5: We appreciate the opportunity to clarify this point. The prevalence period for the CMDQ referred to the preceding workweek. The questionnaire was administered on random days of the week (weekdays) based on availability of participants (refer response to Reviewer1-2). Recruitment mostly occurred in the mornings when participants were more inclined to participate (as opposed to while occupied with work). However, irrespective of the day or time of day that the questionnaire was administered, the participant was instructed to use the week prior as reference starting from Monday morning. We had stated on this in the subsequent paragraph, “Participants were instructed to use the full workweek prior to when the questionnaire was presented as the reference period (i.e., a 7-day period starting Monday morning)” (lines 179-181, Section 2.2. Cornell Musculoskeletal Discomfort Questionnaire).

Reviewer1-6: L197    Do prior publications of aggregate CMDQ score use the term ‘pain score’?  If not, you should use the same term used in prior publications.

Response 6: The CMDQ considers aches, pain and discomfort together, and does not distinguish between these terms. There is no standard term for the score. The developers of the CMDQ simply referred to it as weighted score. Prior studies have used different terms including weighted score, discomfort score. However, the term weight score is not sufficiently clear. The term “discomfort score” could be confused with “discomfort rating”. To avoid potential confusion, we opted to use the term “pain score” to distinguish it from the discomfort ratings. We made the following change in the Methods section to explain this, “the CMDQ provides a procedure for obtaining an aggregate score (which we term ‘pain score’ to differentiate from the term ‘discomfort rating’)” (line 193-194).

Reviewer1-7: L201    It is not clear how the summation facilitates comparisons since the number of body parts (18) is different for each of the 4 body regions. Therefore, if pain is the same in each body part, the pain scores will be larger in the regions with the most body parts.

Response7: We appreciate the opportunity to clarify this. There are two aspects to this. First, the purpose of the four body regions was to facilitate comparisons between job categories (and not between body regions). The statistical analysis illustrates this (i.e., job category as the independent variable, and not body region). Second, the four body regions were constructed based on their anatomical proximity and known similarity in risk factors that affect disorders in these regions. For example, disorders in the wrist, elbow and shoulder have some similar risk factors in the ergonomics literature such as repetitive motions and force exertions involving the upper extremities (and not lower extremities). Disorders in the ankle, knee and hip joints are associated with walking, cart pushing-pulling, prolonged low sitting/squatting, etc. Thus, from an ergonomics perspective such groupings strike a balance between the “whole body” score vs. the very granular individual body parts. We hope the reviewer agrees.

Reviewer1-8: L206    Analysis.  The inclusion of a worker’s body part with “slight” discomfort in the analyses degrades the importance of this report.  Almost everyone has “slight” discomfort throughout the work week – including this reviewer and likely every co-author of this paper.  It would be much more impactful to include only those with moderate or very uncomfortable discomfort/pain and those with substantial interference with work. [And those taking medications for the discomfort, which was not assessed].  This is somewhat dealt in Section 3.4 by reporting the “pain score” that using weighting to discount “slight”.

Response 8: This study was about the 1-week period prevalence of aches, pains and discomfort that are work-related (- the CMDQ does not distinguish between the terms aches, pains and discomfort). The epidemiological definition of period prevalence is the proportion of persons with a particular disease or attribute at any time during the interval under consideration. While we understand the reviewer’s viewpoint, changing the criteria or threshold would result in a deviation from the standard definition for “prevalence” and only cause further confusion. On a related note, we have provided detailed breakdown of counts by each level of discomfort frequency (and severity, work interference) as supplementary data, so the data to compute proportions by “moderate’ or “very uncomfortable” is available to readers.

Reviewer1-9: L212    Was there a formal statistical test for normality of each of the outcome measures?

Response 9: The intent of this question was not entirely clear to us. The test for normality was only relevant for a few of the outcome measures that were continuous variables (e.g., demographic variables and pain scores), and not counts or proportions. We referred to the Kolmogorov-Smirnov test and Shapiro-Wilks test for normality, however these tests are very conservative. We also looked at QQ-plots to check for visual deviations from normality. The demographic variables of age and ‘years on the job’ met conditions of normality (and homoscedasticity) hence ANOVA was considered appropriate. ‘Hours on the job’ showed slight deviation from normality. However, a non-parametric Kruskal-Wallis test and parametric ANOVA produced a similar result of no difference among job categories. Hence, we retained the ANOVA result as-is. The pain scores were non-normal for some of the body regions. To avoid any issues, we opted for the non-parametric Kruskal-Wallis test for all five analyses related to pain scores. Since we used non-parametric tests, we did not consider it necessary to provide details about normality testing.

We hope we correctly interpreted the reviewer’s comment, else respectfully seek additional clarification.

Reviewer1-10: The paper alternates between using SD and SE.  Please only use SD.

Response 10: It is typical for statistical software (SPSS in our case) to present standard errors (SE) when presenting mean differences (e.g., for pairwise comparisons, Chi-square tests, Kruskal-Wallis tests, etc.). Hence we have opted to use SE throughout the manuscript in interest of consistency. We converted SD to SE in Table 1 reporting on age, years of work experience, and hours worked per day. If needed, the SD in Table 1 can be easily computed as follows, SD = SE x sqroot(n). We hope this addresses the issue of consistency.

Reviewer1-11: L246    remove “(0.02 years)”

Response 11:  We have made the deletion as requested.

Reviewer1-12: L259    Overall, results are reported with too many significant digits.

Response 12: In this paragraph, we have reduced the number of significant digits for the ExpB (95% CI) down to two significant digits. We have retained the p-values at three significant digits based on recommended practice/convention and consistency throughout the manuscript.

Reviewer1-13: L259    It is not clear what the value is of repeating in the text all the numbers that are already in the table.  The text should be reduced.

Response 13: We would like to note that not all the numbers in the text are in Table 2. Table 2 provides comparisons for one model, i.e., collectors (vs. reference group; RG), dismantlers (vs. RG), burners (vs. RG) – but does not consider comparisons between dismantlers vs. collectors), burners vs. collectors, and burners vs. dismantlers. These require separate models (i.e., running separate models with a different reference category), and are only described in the text to avoid multiple tables. Other comparisons in the text only take up 1-2 sentences and are provided for completeness. For these reasons, we have retained the text as is.

Reviewer1-14: L284    Fix the insert (Figure 2)

Response 14: Thank you for drawing our attention to the error. This has been corrected and the phrase “Figure 2” has been inserted.

Reviewer1-15: L287    Perhaps in the text just present the findings of significantly different body regions: knee, leg, upper arm.

Response 15: We appreciate the feedback and agree that some text could be reduced. The statistics on e-waste combined vs. reference group are relevant for comparisons with other studies that do not separate by job category – hence we have retained that one sentence. We have made the following changes/deletions to the paragraph in Section 3.3. Prevalence of Musculoskeletal Discomfort: 

Overall discomfort prevalence ± SE in at least one body part was significantly higher for e-waste collectors (91.8 ± 3.2%) compared to the reference group (70.7 ± 7.1%; p < 0.05), though paired comparisons involving dismantlers (89.0 ± 3.5%) and burners (81.0 ± 8.6%) were not significantly different despite their relatively high prevalence compared to the reference group. Figure 1 provides a graphical summary of discomfort prevalence by body part and job category. Prevalence statistics for the right and left sides of the upper and lower extremities were similar; hence, only data for the right side are presented. Comparing across body parts, discomfort prevalence for e-waste workers (vs. the reference group) was highest in the lower back (65.9% vs. 51.2%), followed by the shoulders (37.5% vs. 31.7%), knees (37.5% vs. 19.5%), lower legs (26.7% vs. 14.6%), upper arms (28.4% vs. 2.4%) and neck (26.1% vs. 22.0%). Chi-square tests of proportions indicated statistically significant differences in discomfort prevalence by job category for the knees (p = 0.001), lower legs (p < 0.001), and upper arms (p < 0.001), with the higher discomfort prevalence among collectors driving most of these differences. Prevalence of lower back discomfort was slightly higher among collectors (67%) and dismantlers (68%) compared to burners (52%) and the reference group (51%), however these differences were not statistically significant (p = 0.173). Discomfort prevalence for the shoulders ranged between 29 – 42% (p = 0.599) and for the neck between 22 – 28% (p = 0.894) across job categories. Discomfort prevalence for the remaining body parts were below 20% and did not differ significantly among job categories.”

We hope the reviewer finds these changes satisfactory.

Reviewer1-16: Figure 1: The order of jobs in the Job Category box should be reversed to match the figure.

Response 16: Thank you for pointing this out. We have inserted a new figure with the key re-ordered to match the bar graph.

Reviewer1-17: L309    Fix the insert.

Response 17: Thank you for drawing our attention to the error. This has been corrected and “Figure 2” has been inserted.

Reviewer1-18: Figure 2 is useful and the box insert is correctly ordered.

Response 18: We thank the reviewer for the positive feedback.

Reviewer1-19: L378    change to “165 population studies”; change SE to SD; change “pain/discomfort” to “pain” to accurately reflect the findings from reference 71.

Response 19: We thank the reviewer for our inadvertent error. We have changed “165 studies” to “165 population studies”. We have changed “SE” to “SD” and “pain/discomfort” to “pain” to reflect the findings from reference 71.

Reviewer1-20: L380    Ref 71 reviewed studies of low back “pain”.  Your study evaluated low back “ache, pain, discomfort”.  Therefore, one would expect the prevalence in your study to be higher.

Response 20: We are not sure about the interpretation of this comment. We can expect prevalence to be higher/lower for a variety of reasons, e.g., difference in prevalence period, definition of cases, measurement methods (self-reported vs. clinical diagnoses, etc.). In the second sentence of this same paragraph, we stated “The differences in prevalence periods and questionnaire wording suggest caution when comparing prevalence rates across different studies”. Our intent here is only to point out the low back pain is a work-related issue more globally, and could also be a concern for this e-waste population. We do not claim that the e-waste population is more or less important than other worker populations. If we have misinterpreted the reviewer’s comment, then we respectfully seek further clarification.

Reviewer1-21: L465    The lower age in the e-waste group compared to the reference group is a modifier and one would expect a lower prevalence of discomfort in the e-waste group.  This should be mentioned.

Response 21: We generally agree that this might be the expectation, but the findings suggests otherwise. To address the reviewers comment we added the following to the sentence “The e-waste workers at this site were relatively young …. with relatively few years of experience in e-waste recycling …. which may lower expectations of MSD symptom prevalence compared to the reference group” (line 443-444). Please note that age, a factor contributing to MSDs, is secondary to exposure (intensity, duration, and limited rest/recovery). Also, high physical exposures at a young age before adulthood could have more deleterious effects. Hence, it is not surprising to observe a higher prevalence in the group performing the most demanding tasks for an equivalent number of days and hours of work between the two group. That being said, the mean age of the reference group is only 31 years which is still considered as the age range of young adults.

Reviewer 2 Report

Page 4, Line 163: How was the primary job category determined? Was the time proportion used in each job category? Was the last week considered?

Page 5, Line 199: Using different scales for each rating would give more importance to one factor than another (for example, frequency). Why wasn't a standardized scale used to find the pain score?

Author Response

Reviewer2-1: Page 4, Line 163: How was the primary job category determined? Was the time proportion used in each job category? Was the last week considered?

Response 1: We thank the reviewer for reviewing our manuscript. We appreciate the opportunity to elaborate on this aspect of the study. The primary job category was based on self-reported information. For e-waste workers, this was information provided in response to questions about the tasks they performed most often (i.e., the majority of their work time) including in the past workweek. The classification was relatively straightforward for the majority of participants with the interviewer and participant quickly settling on the job category that was most appropriate. For a few participants, a primary and secondary job category were obtained. We had stated this in the Methods (lines 156 – 161) and Discussion (lines 533 – 537). For further clarification, we added the following phrase to the Methods section: “In such cases, follow-up questions were asked emphasizing tasks performed most often in the prior workweek in order to record the primary and secondary job categories that the participant self-identified with” (lines 155-157).

A few related points are worth noting. While the interview occurred in English, our trained interviewers also offered translations in the local dialect when needed to minimize misinterpretation (lines 166-167). The over 10+ human subject studies (on different environmental and occupational health topics) conducted by our team/colleagues with e-waste workers at Agbogbloshie under the GEOHealth project consortium use this same general procedure [22, 25, 32]. Our interviewers have conducted interviews on multiple of these studies. Hence, we are confident about the consistency in administering the interviews and questionnaires. However, the potential for misclassification is always present when relying on self-reported information in informal work settings. We acknowledge this issue in Discussion, Section Study Limitations:

“Since workers at this site did not have assigned job titles or designations, job categories were derived from self-reported work descriptions. This could cause potential misclassification of the primary job category.” (lines 530 – 532).

We hope this addresses the reviewer’s concerns.

Reviewer2-2: Page 5, Line 199: Using different scales for each rating would give more importance to one factor than another (for example, frequency). Why wasn't a standardized scale used to find the pain score?

Response 2: We are not sure if we understand this comment. The questionnaire that was the used (CMDQ) and the scoring (i.e., weights and procedure used for the pain score) is based on a standardized methodology – and not our own. We have provided references for the CMDQ [54,60,61] and citations for other studies that used this questionnaire methodology and scoring procedure [62-68]. In fact, it is typical for MSD symptom questionnaires to use multiple sub-scales rather than a single scale given the multidimensional nature of MSD symptoms (e.g. frequency, severity, and extent for work interference). We respectfully seek further clarification if we have misinterpreted the reviewer comment.

Reviewer 3 Report

It's a captivating experience reading through this article by learning the astonishing fact that the volume of the e-waste and manual processing of the recycling have brought profound environmental and health crisis for developing countries and local workers. The same question kept pounding me while reading the article, "what can we do to solve the crisis".

Hopefully this research finding will draw further scientific investigations and call global collaborations between countries and manufacture and recycling chains to solve the crisis. 

The authors did a wonderful job to address every perspective of the study. The research background is well versed. The results are well presented. It's interesting to learn how the survey was delivered via local buy-in and facilitation.

The analysis and discussion are sound and logical backed up by findings and comparisons. The conclusions provide as a powerful voice for more attentions, further research and a solution! 

A question for Line 284. “Error! Preference...”. Could the authors explain and fix it?  

Author Response

Response to Reviewer 3 Comments

Reviewer3-1: It's a captivating experience reading through this article by learning the astonishing fact that the volume of the e-waste and manual processing of the recycling have brought profound environmental and health crisis for developing countries and local workers. The same question kept pounding me while reading the article, "what can we do to solve the crisis".

Hopefully this research finding will draw further scientific investigations and call global collaborations between countries and manufacture and recycling chains to solve the crisis. 

The authors did a wonderful job to address every perspective of the study. The research background is well versed. The results are well presented. It's interesting to learn how the survey was delivered via local buy-in and facilitation.

The analysis and discussion are sound and logical backed up by findings and comparisons. The conclusions provide as a powerful voice for more attentions, further research and a solution! 

Response 1: We thank the reviewer for the positive response and feedback about the study. We wholeheartedly agree about the need for collaboration, and hope our study will help advance the dialog.

Reviewer3-2: A question for Line 284. “Error! Preference...”. Could the authors explain and fix it?  

Response 2: Thank you for drawing our attention to the error. The text in question should have read “Figure 1”. The embedded links to Figure 1 caused an error when converting to the journal template. We have rectified the error.

Reviewer 4 Report

Thank you for allowing me to review the article entitled: "Musculoskeletal Disorder Symptoms among Workers at an Informal Electronic-Waste Recycling Site in Agbogbloshie, Ghana".

The prevalence of musculoskeletal disorders must be studied in all possible work, in order to establish prevention measures, so the paucity of studies in the type of work referred to in the article would be the main novelty of the research. I am grateful for the opportunity to know more about this type of work, since I found the article a very pleasant and interesting read.

Abstract:

The objective of the study and the methodology are missing in the abstract

Introduction

The introduction seems to be comprehensive and explain perfectly the magnitude of the problem, thanks to the authors for making me think about this serious matter.

Methods

I think that the main limitation of the study, and that it seriously compromises the objective of comparing the symptoms of discomfort, pain and discomfort of WRMSD among three categories of workers (i.e. collectors, dismantlers and burners) at an informal e-waste recycling site in Agbogbloshie, Ghana, and a reference group of workers who were not involved in e-waste recycling, is that the comparison group cannot be comparable because the jobs they do are totally different, I think the results of this comparison are not useful.

I believe that the objective of examining the prevalence of musculoskeletal disorders among the three types of workers at the e-waste plant was met, and that the results are interesting and can lead to a discussion about the risks of each of the three jobs and the most appropriate profile for each of them.

As for the magnitude of the musculoskeletal disorders, if this measure is based only on the amount of pain, this perception being so subjective and dependent on psychosocial factors, it could be that this objective is not reached with the design made either.

Results

They are well structured and described, although it was expected that the comparison group would have less prevalence of musculoskeletal disorders. In my view, the most striking result from a developed country's perspective is the youthfulness of the workers and the prevalence of pain in such young people. Perhaps a study of mixed quantitative-qualitative methodology with semi-structured interviews or some focus group would be of interest to know not only the prevalence of problems but also the multiple psychosocial causes.

My suggestion is that the authors try to publish a descriptive study of the data concerning a comparison between the three groups of workers in the waste plant, a job they have practically done, and forget the objective of comparing with another reference group and obtaining results on the magnitude.

Conclusions

The main conclusion of the study would be to know something that could be assumed since the groups are not comparable, that e-waste workers are more prevalent than the control group. The following conclusions regarding prevalence are correct and I encourage the authors to publish them again since the problem generated is of everyone.

Author Response

Response to Reviewer 4 Comments

Reviewer4-1: Thank you for allowing me to review the article entitled: "Musculoskeletal Disorder Symptoms among Workers at an Informal Electronic-Waste Recycling Site in Agbogbloshie, Ghana".

The prevalence of musculoskeletal disorders must be studied in all possible work, in order to establish prevention measures, so the paucity of studies in the type of work referred to in the article would be the main novelty of the research. I am grateful for the opportunity to know more about this type of work, since I found the article a very pleasant and interesting read.

Response 1: Thank you for reviewing our article. We agree with the reviewer’s comment about the need for such research, particularly on workers in informal work settings. This was one of the motivations for the present study. We appreciate the feedback and suggestions provided. We have addressed these in the point-by-point basis in our response below.

Abstract:

Reviewer4-2: The objective of the study and the methodology are missing in the abstract

Response 2: Thank you for the feedback. This journal requires a short abstract of about 200 words hence we tried to be concise. The objective of the study is reflected in lines 19 -22 as follows: “This survey study examined the prevalence and magnitude of musculoskeletal disorder (MSD) symptoms among 176 e-waste workers (73 collectors, 82 dismantlers and 21 burners) at Agbogbloshie in Ghana—the largest informal e-waste dumpsite in West Africa – relative to a group of 41 workers not engaged in e-waste recycling.”

To summarize the methodology, we added following sentence to the Abstract, “Study participants were administered a standardized musculoskeletal discomfort questionnaire.” (lines 22- 23). We hope this addresses the reviewers concern.

Introduction

Reviewer4-3: The introduction seems to be comprehensive and explain perfectly the magnitude of the problem, thanks to the authors for making me think about this serious matter.

Response 3: Thank you for the positive feedback.

Methods

Reviewer4-4:  I think that the main limitation of the study, and that it seriously compromises the objective of comparing the symptoms of discomfort, pain and discomfort of WRMSD among three categories of workers (i.e. collectors, dismantlers and burners) at an informal e-waste recycling site in Agbogbloshie, Ghana, and a reference group of workers who were not involved in e-waste recycling, is that the comparison group cannot be comparable because the jobs they do are totally different, I think the results of this comparison are not useful.

I believe that the objective of examining the prevalence of musculoskeletal disorders among the three types of workers at the e-waste plant was met, and that the results are interesting and can lead to a discussion about the risks of each of the three jobs and the most appropriate profile for each of them.

As for the magnitude of the musculoskeletal disorders, if this measure is based only on the amount of pain, this perception being so subjective and dependent on psychosocial factors, it could be that this objective is not reached with the design made either.

Response 4: We appreciate the feedback and the opportunity to elaborate on these aspects of the study. We discuss this in three parts, the use of a reference group, self-report discomfort questionnaires, and psychosocial factors.

Reference group: We included a reference group for context, because we otherwise would not know if the MSD symptom prevalence among e-waste workers at this site is specific to this population or common/endemic to the local area. Many readers will not be familiar with the local conditions in Accra, Ghana (including us at the onset). Hence, we compare the MSD symptom prevalence in this e-waste work population to a somewhat random mix of workers that are not engaged in e-waste work but have similar demographics as the e-waste worker population (e.g., similar ethnicity, gender, age, local environmental/geographical conditions, presumably diet, education level, etc.) – collectively labelled as the “reference group”. The inclusion of such a reference group is not a novel approach in ergonomics studies. Some studies compare between different locations but the same occupation [42], others compare between different occupations in the same area/location [41]. We think such comparisons add value – as long as the data collection methods are comparable. Clearly our findings provide some indication that certain MSD symptoms may be more commonplace in the area (e.g., low back) and not limited to just e-waste workers. That being said, the risk factors may be very different (e.g., low back disorders could be caused from prolonged sitting in sedentary jobs vs. walking/pushing/material handling in e-waste work), though we have avoided such comparative statements about risk factors since it was not the focus of the study. We compare our findings to other studies on e-waste workers conducted on other countries also for context – but of a different nature.

Self-report MSD symptom questionnaires: In our opinion, the reviewer’s concerns are with the methodology more generally and not limited to just our study. There is a vast body of ergonomics literature that has relied on self-report MSD symptom questionnaires as a screening tool for MSD symptoms – including in studies published in this same journal (IJERPH). These are validated and standardized questionnaires with established psychometric properties. We provided references specific to the widespread use of the CMDQ in the Methods section, stating “The CMDQ was used to obtain information about MSD symptoms (e.g., discomfort, aches and pains) experienced in the previous 7-day workweek [54,60,61]. Adapted from the Nordic Musculoskeletal Questionnaire [56], the CMDQ is a widely used screening tool for musculoskeletal discomfort complaints with established psychometric properties applied to diverse occupations [54,62-65] and different language translations [66-68].” (lines 170-174). We also explain in the Discussion that “Self-report questionnaires are widely used as a reliable and cost- and time-efficient method to screen for MSD symptoms in ergonomics studies [58]” (lines 496 – 498).

Psychosocial factors: The local context, the psychosocial factors, and cultural factors all contribute to people’s perceptions about their conditions, including health status. We have devoted an entire section 4.3 Sampling Considerations discussing the specific work context at Agbogbloshie that may be contributing to the high self-reported prevalence of MSD symptoms. We specifically call out the psychosocial factors/stressors with references [22,23,34,39] (lines 476 – 483). These conditions are inherent to an unregulated and unstructured work context, with mostly low wage workers for whom this is the only means of survival and supporting their families. Such conditions may cause many e-waste workers to underreport their problems since they feel resigned to their fate, i.e., “a low sense of agency in influencing their own work situation”. Even more reason for us to be raising awareness about their situation.

We are at a loss about how else we could have addressed the reviewers concern and are open to suggestions.

Results

Reviewer4-5: They are well structured and described, although it was expected that the comparison group would have less prevalence of musculoskeletal disorders. In my view, the most striking result from a developed country's perspective is the youthfulness of the workers and the prevalence of pain in such young people. Perhaps a study of mixed quantitative-qualitative methodology with semi-structured interviews or some focus group would be of interest to know not only the prevalence of problems but also the multiple psychosocial causes.

My suggestion is that the authors try to publish a descriptive study of the data concerning a comparison between the three groups of workers in the waste plant, a job they have practically done, and forget the objective of comparing with another reference group and obtaining results on the magnitude.

Response 5: Thank you for the comment. Prior studies have provided descriptive, qualitative accounts about the work conditions and psychosocial factors present at this informal e-waste recycling site (-- and not waste plant, which may suggest a more formal setting). We cited these in the manuscript [e.g., 22, 23, 26, 28]. It is the lack of quantitative data using standardized screening tools that provided a motivation for this study. We made the case for this in the Introduction,

“Prior studies conducted at Agbogbloshie provide some indication of WRMSD symptoms among e-waste workers. These include descriptive accounts of chronic body pain and discomfort and an overuse of pain medications [22,23,34]. A recent survey of 84 e-waste workers at Agbogbloshie by Fischer et al. found a high prevalence of back pain (88% vs. 69.9% among non-e-waste workers) and neck pain (44.6% vs. 43% among non-e-waste workers) [41]. Studies on e-waste workers in other countries suggest similar trends [11,42,52]. These prior studies focused only on a few and notably different body parts with no data on musculoskeletal discomfort in other body parts, which limits a full understanding of the potential risk for developing MSDs. The lack of a standardized questionnaire methodology and resulting diversity in prevalence periods and symptom definitions hampers comparisons of MSD symptom prevalence across studies. Ergonomics research, on the other hand, has relied on standardized musculoskeletal discomfort questionnaires to screen for WRMSDs [53-58]. The use of such validated questionnaires enable comparisons of MSD symptoms across countries, cultures, work settings, and time characteristics.” (lines 105 – 116).

We hope the reviewer agrees with our point of view. Else we respectfully seek further clarification and/or suggestions for improvement.

Conclusions

Reviewer4-6: The main conclusion of the study would be to know something that could be assumed since the groups are not comparable, that e-waste workers are more prevalent than the control group. The following conclusions regarding prevalence are correct and I encourage the authors to publish them again since the problem generated is of everyone.

Response 6: We are not entirely clear about the intent of this comment. We have purposely avoided the use of the term “control group” because we are not controlling for physical exposures in any way. We have explained above (please see our response to the comment Reviewer4-4) our rationale for using a reference group. We have attempted to present our findings in a detailed manner such that it allow readers to examine or ignore specific between-group comparisons without detracting from the main purpose of the study. We hope the reviewer agrees.

Reviewer 5 Report

This study used a cross-sectional survey to examine the prevalence and magnitude of musculoskeletal disorder symptoms among e-waste workers at Agbogbloshie in Ghana. This study has the valuable implications for future interventions on preventing MDS for this occupation. This paper is well written and presented. I have very few minor comments as below:

  1. The study recruited 176 e-waste workers. Just wondering how many initial recruitment had been done, and what was the response rate for the survey?
  2. A Poisson regression was performed to estimate the number of body parts adjusting for other co-variates. It would also be interesting to see what factors are associated with pain scores such age, years to this job, hours of work per day and occupation.
  3. The study had a reference group for comparison. In the discussion, the author might can compare the prevalence with occupations which are also heavy labour involved such as construction workers.
  4. In the study implication section, the author can also think about what can learn for developed countries. For instance, developing a protocol for manual handing. And I noticed that the age of collectors ranged from 11 to 43, therefore, the occupational health safety for very young worker is also need to be focused.

Author Response

Response to Reviewer 5 Comments

This study used a cross-sectional survey to examine the prevalence and magnitude of musculoskeletal disorder symptoms among e-waste workers at Agbogbloshie in Ghana. This study has the valuable implications for future interventions on preventing MDS for this occupation. This paper is well written and presented. I have very few minor comments as below:

Reviewer5-1: The study recruited 176 e-waste workers. Just wondering how many initial recruitments had been done, and what was the response rate for the survey?

Response 1: Thank you for reviewing our work and providing us with positive feedback. This study used a convenience sampling method. We make this clearer in our revised manuscript (lines128-129; 147). We are unclear about the term “initial recruitments”. The study recruited 176 e-waste workers based on their availability and willingness to participate in the study. During the study period, participants were recruited by approaching workers in-person, the study purpose was explained, and interested workers were invited to participate in the study. Only volunteers participated. The number of persons that either declined to participate or were uninterested were not counted. Hence are unable to quantify a response rate. We acknowledge the limitation of such self-selection in Section 4.4. Study Limitations.

Reviewer5-2: A Poisson regression was performed to estimate the number of body parts adjusting for other co-variates. It would also be interesting to see what factors are associated with pain scores such age, years to this job, hours of work per day and occupation.

Response 2: Thank you for the feedback. We limited our use of Poisson regression to “number of body parts affected” since it was a single aggregate outcome measure. All of the other analysis were done by body part and/or body region. In our view, the analysis of pain scores by body regions is equally if not more informative than whole body pain scores since (1) even if the whole body pain scores are equal, the body regions contributing to it could be very different and so body regions would be a confounder – which limits its usefulness, and (2) the body regions were intentionally grouped this way because it gives us some indication of what could be potential work-related risk factors (i.e., prior research informs us that the risk factors that affects upper extremities are quite different from factors affecting lower extremities). Performing Poisson regressions for each of these body part prevalence and/or region would be repetitive, not to mention tedious for readers. We hope the reviewer agrees.

Reviewer5-3: The study had a reference group for comparison. In the discussion, the author might can compare the prevalence with occupations which are also heavy labour involved such as construction workers.

Response 3: We appreciate the opportunity to elaborate on this aspect. Many occupations involve heavy labor including construction – and so any such discussion need not be limited to construction. However, such comparisons would be beyond the scope of this study and potentially detracting from our main objective. Rather we have focused our comparisons in the Discussion on two aspects:

(1) MSD symptom prevalence in this e-waste work population compared to a somewhat random mix of workers not engaged in e-waste work, but have similar age, ethnicity, gender, environment/geography, etc. (i.e., labelled as the “reference group”). We did this to provide context since many readers will not be familiar with the local conditions in Accra, Ghana. The inclusion of such a reference group is not a novel approach. We think it adds value. Clearly our findings provide some indication that certain MSD symptoms may be more commonplace in the area (e.g., low back) and not limited to just e-waste workers. That being said, the risk factors may be very different (e.g., low back disorders could be caused from prolonged sitting in sedentary jobs vs. walking/pushing/material handling in e-waste work), though we have avoided such comparative statements about risk factors since it was not the focus of the study.

(2) We compare our findings to other published studies on informal waste workers, specifically e-waste workers, to provide context about whether our findings are limited to this one worksite or are part of a larger trend in informal e-waste processing occurring globally.

We have tried to be as thorough and clear in our presentation of findings. This level of detail should allow readers to make their own comparisons with other worker populations and/or work settings, if needed. We hope the reviewer agrees.

Reviewer5-4: In the study implication section, the author can also think about what can learn for developed countries. For instance, developing a protocol for manual handing. And I noticed that the age of collectors ranged from 11 to 43, therefore, the occupational health safety for very young worker is also need to be focused.

Response 4: We agree that ergonomic interventions, such as safe material handling practices, etc. are needed. However, we are also mindful of the limitations of our cross-sectional study and absence of detailed information about the physical risk factors associated with the work-related disorder symptoms. We acknowledge these points in Section 4.4. Study Limitations. Our on-going research focuses on associations between ergonomics risk factors and MSDs, and subsequent development, implementation and evaluation of interventions. The reviewer clearly sees value in this research and hence we find the feedback comment to be encouraging.

Round 2

Reviewer 4 Report

I am grateful for the authors' point-by-point responses.

As for the abstract, in it the authors indicate what they did: This survey study examined the prevalence and magnitude of...... I was referring to put it in a formal way: The objective of this study was to examine the prevalence and..... to evaluate whether to change it or leave it as it is.

Otherwise, all ok

As for my comment about comparing groups, I still think that to know if the type of work produces more prevalence of musculoskeletal disorders, you cannot compare 176 e-waste workers with only 41 shop attendants (n = 8), or traders (n = 20), or vehicle drivers (n = 4), or students (n = 3) or a couple each of schoolteachers, or tailors, and unemployed youth, many of whom did not typically perform heavy force exertions or MMH as part of their routine work. I believe they are not comparable. I that because of this the results are entirely to be expected.

In the study (42) to which the authors refer, the groups are comparable: "The objective of this study was to complete comprehensive health evaluations on e-waste recycling workers in Chile and to compare those that work in informal (i.e., independent) to those that work in formal (i.e., established company) settings", and attempts to find out whether or not working for a company is a determining factor. As for the study (41), the first difference with the present one is that the number of participants in each of our groups was similar, n=84 and n=94, and the second is that it is specified that most of them performed similar jobs or at least in the same environment as the e-waste workers: "Onion carriers, for example, loaded onion sacks from and onto trucks by the main road, scraps traders provided workers with devices, and sellers crossed the scrap yard daily with their goods coming from shops at the periphery of the yard". I think that at least the possibility that the groups were not comparable should be taken into account as one of the major limitations of the study.

On the other hand, I have no problem with self-report symptom questionnaires, which I myself have used in some studies. My doubt concerns the use of the measure "magnitude". Are the authors really measuring the magnitude of the problem? Recall that the word magnitude refers to determining the "size of the problem", how big and important a health problem is, and they do this through a scale that measures frequency, severity and interference with work, the 'discomfort rating', which becomes a 'pain score'. I think a proposed objective, and one that the authors should consider for consistency with the results of the study, would be: to examine the prevalence, perceived intensity and its interaction with work capacity of musculoskeletal disorder (MSD) symptoms among 176 e-waste workers (73 collectors, 82 dismantlers and 21 burners) at Agbogbloshie in Ghana - the largest informal e-waste dumpsite in West Africa - relative to a group of 41 workers not engaged in e-waste recycling. This was my concern. This was my concern. Perhaps not being familiar with the scale makes me not objective enough, but please consider my assessment.

As for the conclusions, my comment was still focused on the fact that these two groups cannot be compared, and that due to the nature of the groups, the main conclusion "Overall, the musculoskeletal discomfort frequency (i.e., 1-week period prevalence) and magnitude (i.e., weighted pain scores) was higher among e-waste workers compared to the reference group", was entirely expected.

I insist that perhaps this could be remedied by including the possibility that the groups were not comparable as one of the major limitations of the study.

I hope I have clarified my comments and that they will be helpful to the authors.

Author Response

Response to Feedback Comments from Reviewer-4

Reviewer: I am grateful for the authors' point-by-point responses.

Reviewer-4-1: As for the abstract, in it the authors indicate what they did: This survey study examined the prevalence and magnitude of...... I was referring to put it in a formal way: The objective of this study was to examine the prevalence and..... to evaluate whether to change it or leave it as it is.

Otherwise, all ok

Response 1: We thank the reviewer for the comments and suggestions. We have revised the beginning of this sentence as follows: “The aim of this study was to examine the …..”  (Line 19). We have made other edits to the aims sentence in response to comment R4-3. Please see below.

Reviewer-4-2: As for my comment about comparing groups, I still think that to know if the type of work produces more prevalence of musculoskeletal disorders, you cannot compare 176 e-waste workers with only 41 shop attendants (n = 8), or traders (n = 20), or vehicle drivers (n = 4), or students (n = 3) or a couple each of schoolteachers, or tailors, and unemployed youth, many of whom did not typically perform heavy force exertions or MMH as part of their routine work. I believe they are not comparable. I that because of this the results are entirely to be expected.

In the study (42) to which the authors refer, the groups are comparable: "The objective of this study was to complete comprehensive health evaluations on e-waste recycling workers in Chile and to compare those that work in informal (i.e., independent) to those that work in formal (i.e., established company) settings", and attempts to find out whether or not working for a company is a determining factor. As for the study (41), the first difference with the present one is that the number of participants in each of our groups was similar, n=84 and n=94, and the second is that it is specified that most of them performed similar jobs or at least in the same environment as the e-waste workers: "Onion carriers, for example, loaded onion sacks from and onto trucks by the main road, scraps traders provided workers with devices, and sellers crossed the scrap yard daily with their goods coming from shops at the periphery of the yard". I think that at least the possibility that the groups were not comparable should be taken into account as one of the major limitations of the study.

Response 2: We are grateful for the reviewer’s comment. We agree with the general sentiment expressed. Even in study [42], the worker categories were a random mix of potentially physically demanding work (as pointed out by the reviewer) but also some sedentary work (e.g., tailors, security guards). The term “bystanders” is also potentially misleading and suggests spectators or onlookers. Without knowing the exact work content (type, duration, intensity), we can only speculate whether these two groups were comparable or not.

Regarding sample size, in our study we think the size of the three separate job categories compared to the reference group is what is more relevant, since we are interested in the four-way statistical comparison (3 e-waste plus 1 reference group) and not the two-way comparison between e-waste vs. reference groups (which was mostly the focus of [42]). In our study, burners were the smallest group (i.e., not the reference group), and we described the potential reasons for and implications of this previously in the Discussion (lines 521 – 529).

Our intent of including the reference group was not to prove the obvious, but rather to provide some context to the obtained MSD data, which we explained in the previous round of review.

That being said, we agree that we ought to clarify more explicitly in the Study Limitations that the reference group consisted of workers from diverse occupations and with potentially different work exposures (type, duration, intensity of physical work) compared to the three categories of e-waste workers, which may limit direct comparisons in MSD symptom prevalence and intensity.

To address this, we have made the following additions to Section 4.4. Study Limitations:

“The reference group and e-waste worker cohort recruited were similar in gender and ethnic composition. Inclusion of the reference group helps provide context to the study findings on MSD symptoms among e-waste workers. However, workers in the reference group had diverse occupations, many being less physically demanding, and thus different from informal e-waste recycling in terms of physical work exposure (e.g., task type, duration and intensity). This suggests some caution when drawing conclusions from this study about differences in the prevalence, location and/or intensity of MSD symptom between both groups.” (lines 538 - 544).

We hope the reviewer considers these additions to be satisfactory.

Reviewer-4-3: On the other hand, I have no problem with self-report symptom questionnaires, which I myself have used in some studies. My doubt concerns the use of the measure "magnitude". Are the authors really measuring the magnitude of the problem? Recall that the word magnitude refers to determining the "size of the problem", how big and important a health problem is, and they do this through a scale that measures frequency, severity and interference with work, the 'discomfort rating', which becomes a 'pain score'. I think a proposed objective, and one that the authors should consider for consistency with the results of the study, would be: to examine the prevalence, perceived intensity and its interaction with work capacity of musculoskeletal disorder (MSD) symptoms among 176 e-waste workers (73 collectors, 82 dismantlers and 21 burners) at Agbogbloshie in Ghana - the largest informal e-waste dumpsite in West Africa - relative to a group of 41 workers not engaged in e-waste recycling. This was my concern. This was my concern. Perhaps not being familiar with the scale makes me not objective enough, but please consider my assessment.

Response 3: We thank the reviewer for the feedback. We agree that “intensity” may in fact be a more appropriate term in place of “magnitude” considering the potential for confusion. However, we do not think that “perceived intensity” would be accurate, since the “intensity” measure is computed using self-reported (or perceived) frequency, severity and work interference ratings. Instead we added the term “self-reported” before “musculoskeletal disorder (MSD) symptoms”. We opted to not include the suggested term “work capacity” which has specific meaning/measurements in ergonomics and work physiology and which we did not measure.  Hence, we only focus on the two main outcome measures of “prevalence” and “intensity” – the third being symptom location (i.e., body part) but that is implied when using standardized MSD symptom questionnaires from the ergonomics literature.

With R4-1 and R4-3 combined, the aims sentence was getting too long. So we have revised the Abstract as follows:

“The aim of this study was to examine the prevalence and intensity of self-reported musculoskeletal disorder (MSD) symptoms among e-waste workers at Agbogbloshie in Ghana – the largest informal e-waste dumpsite in West Africa – relative to workers not engaged in e-waste recycling. A standardized musculoskeletal discomfort questionnaire was administered to 176 e-waste workers (73 collectors, 82 dismantlers and 21 burners) and 41 workers in a reference group.” (lines 19 – 23).

Likewise, the aims paragraph at the end of the Introduction section has a similar opening sentence, and reads as follows:

“The aim of this study was to assess the prevalence and intensity of WRMSD symptoms among informal e-waste workers at the Agbogbloshie recycling site using the standardized Cornell Musculoskeletal Discomfort Questionnaire (CMDQ) [54]. The study hypothesized a higher frequency (i.e., 7-day period prevalence) and intensity (i.e., using weighted pain scores) of self-reported musculoskeletal discomfort among the three e-waste worker categories (collectors, dismantlers, and burners) compared to a reference group of workers not engaged in informal e-waste recycling.” (lines 117- 122).

The second sentence in the paragraph above mentions the exact outcome measures, i.e., “frequency (i.e., 7-day period prevalence) and intensity (i.e., using weighted pain scores)” and also indicates “self-reported” musculoskeletal discomfort. In our opinion, the Abstract and Aims paragraph are both consistent.

For consistency, we have also replaced the word “magnitude” with “intensity” in 10 other instances throughout the manuscript (lines, 29, 31, 117, 120, 321, 389, 399, 491, 535, 582).

We hope the reviewer finds these changes to be satisfactory.

Reviewer-4-4: As for the conclusions, my comment was still focused on the fact that these two groups cannot be compared, and that due to the nature of the groups, the main conclusion "Overall, the musculoskeletal discomfort frequency (i.e., 1-week period prevalence) and magnitude (i.e., weighted pain scores) was higher among e-waste workers compared to the reference group", was entirely expected.

I insist that perhaps this could be remedied by including the possibility that the groups were not comparable as one of the major limitations of the study.

I hope I have clarified my comments and that they will be helpful to the authors.

Response 4: We thank the reviewer for this feedback. We hope the revisions made to Section 4.4. Study Limitations (lines 538 -544) in response to Reviewer-4-2 satisfactorily addresses the concern.

We sincerely thank this reviewer for the detailed feedback aimed at improving the clarity of our manuscript. We hope the reviewer finds our revisions to satisfactory and acceptable. Thank you.